# Articulate Anything: Open-vocabulary 3D Articulated Objects Modeling

## Abstract

3D *articulated* objects modeling has long been a challenging problem, since it requires to capture both accurate surface geometries and semantically meaningful and spatially precise structures, parts, and joints. Existing methods heavily depend on training data from a limited set of handcrafted articulated object categories (*e.g.*, cabinets and drawers), which restricts their ability to model a wide range of articulated objects in an open-vocabulary context. To address these limitations, we propose ARTICULATE ANYTHING, an automated framework that is able to convert any rigid 3D mesh into its articulated counterpart in an open-vocabulary manner. Given a 3D mesh, our framework utilizes advanced Vision-Language Models and visual prompting techniques to extract semantic information, allowing for both the segmentation of object parts and the construction of functional joints. Our experiments show that ARTICULATE ANYTHING can generate large-scale, high-quality 3D articulated objects, including tools, toys, mechanical devices, and vehicles, significantly expanding the coverage of existing 3D articulated object datasets. Additionally, we show that these generated assets can facilitate the acquisition of new articulated object manipulation skills in simulation, which can then be transferred to a real robotic system. Our Github website is https://annonymous-user1234.github.io/.

## 1 Introduction

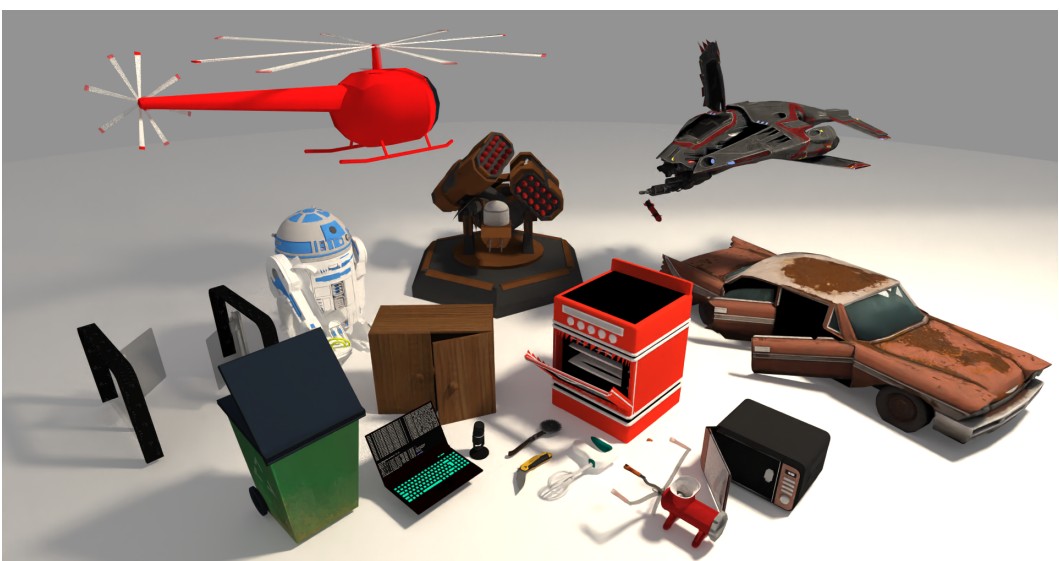

Figure 1: **Articulate Anything** turns 3D meshes (*e.g.* retrieved from Objaverse Deitke et al. (2024)) into articulated objects. Our pipeline is capable of processing a wide range of objects, including daily necessities, furniture, vehicles and even fictional objects.

Gathering data on a large scale is an up-and-coming research trend in embodied AI and robotics. Large foundation models built for robotics and embodied AI are extremely data-hungry, intensifying

the need for large-scale data collection. Compared to collecting data in the real world, gathering data in simulations is significantly faster and more convenient, making it easier to capture various types of ground truth information such as physical states, segmentation masks, depth maps, etc. Existing works have explored many different aspects of how to carry out large-scale data collection in simulations, ranging from asset generation (Chen et al., 2024), scene generation (Yang et al., 2024a;b), task design (Wang et al., 2023), demonstration collection (Dalal et al., 2023; Ha et al., 2023), reward design (Ma et al., 2023), etc.

Despite these efforts, a key challenge remains: generating diverse, realistic articulated objects, which are essential in everyday life, from vehicles to furniture. Successfully generating and simulating these objects is vital for producing data that generalizes to real-world applications. While there has been substantial progress in non-articulated 3D assets generation, few available methods are capable of addressing the demand for generating articulated objects. The majority of 3D generation approaches, utilizing either a forward pass of a trained network (Long et al., 2024; Hong et al., 2023; Xu et al., 2024; Shi et al., 2023a;b), or SDS loss optimization (Poole et al., 2022; Wang et al., 2024; Qiu et al., 2024), produce only the surface of the object. These objects can only be manipulated as a whole body, which often results in functional limitations since all their parts are glued together. For instance, a closet produced through these 3D generation methods cannot be opened or used for storing clothes. Part-aware 3D generation approaches (Gao et al., 2019; Yang et al., 2022; Mo et al., 2019; Wu et al., 2020; Nakayama et al., 2023; Koo et al., 2023; Liu et al., 2024a) generate 3D objects together with their part-specific details. Though these methods are more sensitive to structure, the objects produced are still restricted to whole-object manipulation for the same reason. Articulated object modeling approaches (Jiang et al., 2022b; Liu et al., 2023a; Chen et al., 2024; Lei et al., 2023; Liu et al., 2024c; Mandi et al., 2024; Nie et al., 2023; Wang et al., 2022; Gadre et al., 2021) are capable of producing articulated objects with several interactive parts, demonstrating functionality. However, existing articulated object modeling methods either require dense observation of a to-be-reconstructed articulated object (Jiang et al., 2022b; Liu et al., 2023a; Mandi et al., 2024; Huang et al., 2021), or are restricted on the limited-scale data used to train their network (Chen et al., 2024; Lei et al., 2023; Liu et al., 2024c). As a result, existing approaches for articulated object generation lack the ability to automatically generate a wide variety of articulated objects, especially those from object categories that are either underrepresented or not included in existing articulated object datasets (Xiang et al., 2020; Wang et al., 2019; Geng et al., 2023).

Consequently, the generation of diverse articulated objects presents extra challenges compared to non-articulated 3D assets: (1) it necessitates the generation of accurate semantic structures alongside geometry and appearance, (2) it requires the generation of articulation parameters like joint orientation and position, and (3) the relatively small-scale articulated object datasets, compared to 3D object datasets, are insufficient to support the training of a generalizable generative model. To overcome those challenges, we introduce ARTICULATE ANYTHING, an automated pipeline for converting any 3D mesh into a corresponding articulated asset. In order to go beyond existing articulated object datasets, our pipeline leverages prior knowledge from visual and language foundation models (Achiam et al., 2023; Kirillov et al., 2023; Rombach et al., 2022) and generalizable geometric clues, rather than relying on existing labeled articulated object data. Building upon established 3D generation approaches, our pipeline starts with a 3D mesh, whether generated or handcrafted, followed by the stages of **Movable Part Segmentation**, **Articulation Estimation** and **Refinement**.

In **Movable Part Segmentation**, the goal is to segment out all movable parts and determine their semantics for subsequent articulation estimation. A recent work Part123 Liu et al. (2024a) performs open-vocabulary 3D part segmentation by lifting 2D segmentation masks produced by SAM Kirillov et al. (2023) into 3D. We leverage a method similar to Part123 to generate 3D masks. To additionally acquire the semantics of the generated 3D masks, we use GPT4o to label instance-level semantics on 2D segmentation masks produced by SAM. We prioritize GPT4o over 2D grounding models such as GroundingDINO Liu et al. (2023c) because they encounter an issue that while excelling at instance-level segmentation, they demonstrated reduced capability at part-level segmentation.

After obtaining the 3D segmentation of non-fixed parts and their semantics, the next step in our process involves **Articulation Estimation**. Unlike prior approaches (Li et al., 2020; Huang et al., 2021; Zeng et al., 2024) that rely on a training dataset to predict these parameters, our pipeline achieves articulation estimation in an open-vocabulary fashion. To surpass the limitations of existing datasets, we aim to uncover the inherent geometric clues in articulated objects, rather than relying on learning from pre-existing datasets. Specifically, the joint connecting two segmented parts is

inherently tied to the geometry of the connected area. For example, if the parts are connected in a straight line, like in a laptop hinge or an open door, this line essentially represents the joint. Therefore, we provide GPT-4o with such information through visual prompting skills, allowing it to leverage its common-sense knowledge and the geometric clues to infer the joint parameters. In this way, we can generalize to any unseen objects with accurate segmentation, and the performance will continue to improve as the vision-language model becomes more advanced.

At this stage, we already have a functional articulated object. However, since the articulated objects generated by our pipeline might originate from an object with only surface geometry and texture, the segmented-out 3D parts will inevitably suffer from occlusion, and leave holes in other parts. For example, in the closed state, only the outer surface of the drawer will be segmented, and its internal structure will not be present at all.

Thus, in the **Refinement** stage, we leverage 2D diffusion models (Rombach et al., 2022; Poole et al., 2022) to close holes, enhance the geometry, and improve texture quality. Such models are trained on millions of images, establishing a solid prior about 3D objects, and the 3D prior in them can be easily distilled using SDS (Score Distillation Sampling) loss Poole et al. (2022). While SDS loss is not aware of 3D part segmentation, causing parts to grow into each other, we solve this by a simple yet effective optimization strategy that incorporates random transformations of movable parts according to their joint parameters during the SDS loss optimization phase.

We conducted experiments comparing the unconditional generation capability and joint parameter estimation accuracy against various baselines. Experimental results show that while our method matches state-of-the-art articulated object generation methods for selected categories that they are trained on within PartNet-Mobility dataset Xiang et al. (2020), it also demonstrates the capacity to generate a broader range of articulated objects beyond these categories. We summarize our contributions as follows:

- We introduce ARTICULATE ANYTHING, a pipeline capable of generating diverse, realistic and complex articulated objects in an open-vocabulary manner.
- We propose a novel articulation parameter estimation method that leverages geometric clues and a tailored visual prompting method to estimate joint parameters in an open-vocabulary manner.
- We propose a novel optimization strategy that enables using diffusion prior to refine articulated objects while keeping the semantics of parts from being changed or lost.

## 2 RELATED WORK

### 2.1 ARTICULATED OBJECT MODELING

One line of work focuses on the reconstruction of articulated objects from observations. Jiang et al. (2022b) and Huang et al. (2021) train a network to reconstruct an articulated object from input multi-frame point clouds. Some other works (Hsu et al., 2023; Gadre et al., 2021; Nie et al., 2023; Wang et al., 2022) learn joint parameters from active interaction. Liu et al. (2023a), Weng et al. (2024), Wei et al. (2022) and Mandi et al. (2024) reconstruct the geometry and joint parameters of an articulated object from multiview images obtained at distinct states of the articulated object. Chen et al. (2024) reconstructs articulated objects from a single real-world RGB image through a network trained on large-scale synthetic data. However, these approaches either demand extensive observation of the same object or rely on existing articulated object data to train neural networks, with limited ability to generalize across a small range of categories.

Concurrent with reconstruction approaches, another body of work, including Lei et al. (2023) and Liu et al. (2024c), represents articulated objects as graphs and employs a diffusion model to fit the distribution. The shape, position, and semantic information of the parts are encoded in the vertices, while joint parameters are stored in the edges. These models operate in a generative fashion. Similar to reconstruction methods, these generation methods also depend on existing articulated object datasets to train their diffusion models and are unable to generalize beyond the training data. Overall, none of the existing methods can automatically generate articulated objects in an open-vocabulary manner.

Additionally, a recent survey on articulated object modeling Liu et al. (2024b) covers various topics, including reconstruction, generation, and motion prediction (Jiang et al., 2022a; Li et al., 2020; Hu et al., 2017; Sharf et al., 2014), among others.

## 2.2 PART AWARE 3D GENERATION

Instead of generating an 3D object as a whole, part aware 3D generation methods generate objects with part-level information. Gao et al. (2019) learns two separate VAEs to model global structural information and detailed part geometries. Yang et al. (2022) encodes 3D objects into a representation that disentangles structure and geometry. Mo et al. (2019) utilizes a graph VAE to encode shapes into hierarchical graph representations. Wu et al. (2020) applies a Seq2Seq generative network for part assembly and generation. Nakayama et al. (2023) independently models part geometry and global part transformation, and conditioned on both of them, a diffusion model synthesizes the final shape. Koo et al. (2023), on the other hand, generates part geometry latents with one diffusion model and synthesizes the global geometry with another diffusion model conditioned on these latents. Liu et al. (2024a), a recent release, learns a Neus model from generated multi-view images and corresponding 2D segmentation maps provided by Kirillov et al. (2023), and extracts 3D segmentation masks using a clustering algorithm.

While these methods can produce shapes with plausible structures and part geometries, they frequently depend on object datasets with part-level annotations and fail to generalize beyond the datasets used for training. Furthermore, they do not generate articulation parameters for individual parts, causing the generated parts to be individually non-manipulatable in simulation, which limits their applicability in downstream tasks for robot learning.

## 3 METHOD

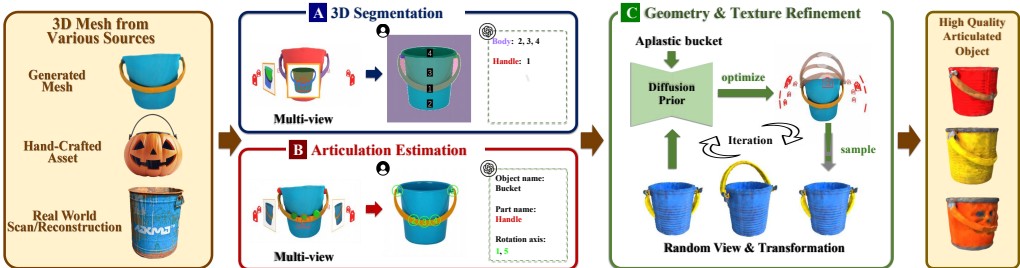

Figure 2: ARTICULATE ANYTHING converts 3D meshes from various sources to high quality articulated objects via three main parts: **A.** Movable Part Segmentation, **B.** Articulation Estimation, **C.** Refinement.

## 3.1 OVERVIEW

Our pipeline converts any mesh into its articulated counterpart, including hand-crafted meshes with part geometry, reconstructed meshes, and meshes generated through text-to-3D or image-to-3D methods. The input mesh undergoes three main steps, including **Movable Part Segmentation**, **Articulation Estimation** and **Refinement**, as depicted in Figure 2. Specifically, the **Movable Part Segmentation** step utilizes SAMKirillov et al. (2023) to extract 2D segmentation masks from object images rendered from multiple viewpoints, analyze the semantics of these masks with GPT4o, and integrate them into 3D masks. **Articulation Estimation** determines the articulation parameters of each movable part by leveraging geometric cues and prior knowledge from visual and language foundation models. In **Refinement** stage, the geometry and texture of the articulated object are enhanced using the geometry-aware 2D diffusion model Richdreamer (Qiu et al., 2024), with SDS loss guiding the distillation of the diffusion model's 3D prior. During this process, movable parts are randomly transformed to improve optimization performance.

## 3.2 MOVABLE PART SEGMENTATION VIA VLM ASSISTANT

In this step, our goal is to segment all movable parts from a 3D mesh and determine their semantics in an open-vocabulary manner. Liu et al. (2024a) segments 3D meshes by lifting 2D segmentation

masks from various viewpoints into 3D space, as illustrated in Figure 2 (A), left image. Specifically, it obtains 2D segmentation masks using Kirillov et al. (2023) and merges their 3D projections when the overlap ratio exceeds a specified threshold.

However, this method often struggles to accurately segment movable parts. For instance, some non-movable parts may be over-segmented (*e.g.*, the front windshield of a car), while certain movable parts may be under-segmented (*e.g.*, two closed drawers). To address these challenges, we propose incorporating a vision-language model to enhance the process. As shown in Figure 2 (A), right image, for each 2D segmentation mask, we employ the Set-of-Mark prompting technique (Yang et al., 2023) to prompt GPT4o (Achiam et al., 2023) to generate captions that describe the movable parts at a semantic level. Based on the semantics provided by GPT4o, we can then merge the masks, resolving the issues observed in Liu et al. (2024a).

### 3.3 ARTICULATION ESTIMATION VIA GEOMETRY-AWARE VISUAL PROMPTING

Given that our goal is to achieve an open-vocabulary scope, estimating articulation parameters relying solely on the limited-scale existing datasets of articulated objects(Mandi et al., 2024; Chen et al., 2024) is insufficient. Instead, we must develop a method that generalizes across a diverse set of articulated objects. To tackle this challenge, we propose a *Geometry-aware Visual Prompting* approach to estimate articulation parameters by exploiting the shared features present in the geometry, mechanical structure and functionality of different articulated objects. Specifically, we observe that joints connecting two parts are strongly influenced by the geometry in areas where the parts are in close proximity, which we define as the *connected area*. To incorporate these geometric cues into GPT4o, we first compute the Signed Distance Field (SDF) for each part and identify a set of points within the connected area. Next, we apply DBSCAN (Schubert et al., 2017) to cluster these points into several key points, as illustrated in Figure 2 (B), left image. Finally, we label the 2D projections of these key points with numbers and prompt the labeled image into GPT4o, guiding it to describe the joint based on these key points, as shown Figure 2 (B), right image.

We primarily focus on two dominant joint types: prismatic joints and revolute joints, and estimate them differently according to their distinct mechanical structures.

**Revolute joints.** Revolute joints connect two parts of an articulated object through physical hinges or other similar hinge-like mechanisms, such as the lid of a cardboard box. The rotation axis of a revolute joint always runs along the length of its corresponding hinges. This implies that identifying the hinges in 3D space allows for a straightforward estimation of the rotation axis. As a hinge physically connects two parts of an object, it is most likely located somewhere within the region where the two parts are connected. From this observation, we estimate the rotation axis of a part by sampling candidate points in the area where the part and the rest of the object are connected. After that, we render an image of the object with those candidate points projected onto the 2D image, and label them using numbers as shown in Figure 2. This image is used to prompt GPT4o to pick two or more points to form the hinge, establishing the rotation axis. In some cases, a hinge may not be positioned on the surface of the object, resulting in only one end being identifiable on the object's surface, which causes only a single point to be selectable (for example knobs). In this case, we assume the direction of the rotation axis to be perpendicular to the plane fitted to the connected area. In this way, the rotation axis is determined by the normal vector of the plane and the one selected point.

**Prismatic joints.** Prismatic joints connect two parts of an articulated object through sliding mechanisms, allowing linear motion along a single axis. However, unlike revolute joints, the sliding mechanism is oftentimes concealed beneath the surface of the articulated object, making it impractical to apply the strategy for revolute joints to prismatic joints.
According to our observation, prismatic joints can be divided into two types based on their functionality: One specific type of prismatic joint enables its child component to slide in and out of the articulated object to which it is connected. Parts featuring this type of prismatic joint include drawers, push buttons, etc. This type of joint can translate in various directions, but to pull a part out of the object, its translation direction must have a large portion directed outward from the object. Therefore, we characterize this type of joint by the normal vector of the plane fitted to the connected area. The other type of prismatic joints allows its child to translate along the surface of the articulated object to which it is attached. Parts featuring this type of prismatic joint include slid-

ing doors, sliding windows, etc. This observation narrows the potential translation direction from three-dimensional to two-dimensional, enabling us to annotate a 2D rendered image with arrows and prompt GPT-4 to select the most appropriate direction. Overall, given an articulated object and its segmented parts, we prompt GPT4o to classify all prismatic joints into the two aforementioned categories, subsequently estimating the translation directions of these prismatic joints based on their category.

### 3.4 Refinement via Randomized Articulation Configuration

After the previous two stages of 3D segmentation and joint estimation, we can already obtain an articulated object with correct structure. However, if the input mesh does not include part geometry, the result is still flawed because object parts are segmented from a mesh containing only surface data, without inner structures. This inevitably leads to some parts being affected by occlusion, resulting in incomplete segments with holes.

Therefore, we incorporated a refinement step into our conversion pipeline. For the purpose of completing occluded geometries, filling holes, and enhancing the rendering quality, we distill 3D object geometry and texture prior from a geometry-aware 2D diffusion model Qiu et al. (2024) into our generated articulated objects using SDS loss Poole et al. (2022). However, directly using existing SDS optimization method Qiu et al. (2024) to optimize the articulated object in a fixed posture leads to inevitable growth and overlap of its parts, resulting in artifacts when the object parts are in motion. To solve this problem, we propose a novel optimization technique for articulated objects, in which we randomly transform the movable parts of an articulated object during the optimization process. In each optimization step, in addition to randomly sampling rendering camera positions and diffusion timesteps, we additionally sample motion parameters (rotation degree and translation length) for each non-fixed part and transform these parts accordingly. This approach ensures that most proportions of the parts are exposed in certain poses, while any overlapping geometry in one pose becomes a noticeable artifact in other poses, enabling it to be gradually optimized. As the object parts are already relatively recognizable at initialization, the diffusion model effectively preserves their semantics, ensuring that the meaning of the parts is rarely altered or lost.

## 4 Experiments

**Implementation** In the 3D segmentation stage, we use semantic-SAM Li et al. (2023) to generate 2D segmentation masks and draw these masks onto 2D images using Set-of-Mark Yang et al. (2023) techniques. In both 3D segmentation and articulation estimation stages, we visually prompt GPT4o for part and structure knowledge of articulated objects. For refinement stage, we implement a multi-part DMTet Shen et al. (2021) for articulated objects representation. Each part of the object is assigned an independent DMTet network for describing its geometry and texture features. We exploit the normal-depth diffusion model and albedo-diffusion model proposed in Richdreamer Qiu et al. (2024) for geometry and texture prior.

**Baselines** To the best of our knowledge, this is the first work to address the challenge of converting meshes into articulated objects in an open-vocabulary manner. Consequently, no existing baseline directly takes the same input and produces the same output as ARTICULATE ANYTHING. Instead, we evaluate individual components of our pipeline by comparing them against suitable baselines for each step. For the 3D segmentation step, we use PartSlip Liu et al. (2023b) and PartDistill Umam et al. (2024) as the baseline. PartSlip leverages the open-vocabulary grounding capability of GLIP and fuses the 2D detections to obtain 3D segmentation. PartDistill utilizes a cross-modal distillation framework to transfer 2D knowledge from VLM for 3D part segmentation. The articulation estimation step in our pipeline takes object parts as input and outputs their joint configurations. NAP and CAGE, which learn diffusion models to generate articulated objects, can also generate joint configurations conditioned on object parts, making them the closest baselines for this step. Additionally, methods such as ANCSH and OPD directly estimate articulation parameters. These approaches take a single observation (e.g., RGB, depth, or both) as input and predict part segmentation and joint parameters simultaneously. We compare these methods to the combined 3D segmentation and articulation estimation steps in ARTICULATE ANYTHING. For the refinement step, since our approach builds on Richdreamer, it serves as a natural baseline for this stage. Lastly, our pipeline also enables open-vocabulary, unconditional articulated object generation by additionally employing a 3D gen-

Table 1: Quantitative evaluation of **3D segmentation**, measured in mIOU (%).

| Method | Categories | | | | | | | | | | | | | | | | Overall↑ |
|---|---|---|---|---|---|---|---|---|---|---|---|---|---|---|---|---|---|
| | Bottle | Chair | Display | Door | Knife | Lamp | StorageFurniture | Table | Camera | Cart | Dispenser | Kettle | KitchenPot | Oven | Suitcase | Toaster | |
| PartSLIP Liu et al. (2023b) | 76.12 | **73.85** | 59.10 | 14.57 | 10.04 | 43.20 | 27.30 | 33.24 | **51.40** | 79.54 | 10.22 | 22.57 | 31.67 | 31.08 | 42.58 | 14.76 | 38.83 |
| PartDistill Umam et al. (2024) | **77.98** | 70.23 | 60.79 | 43.86 | **39.41** | 67.56 | 21.60 | 42.04 | 32.75 | 76.39 | 11.45 | 36.92 | 21.44 | 27.89 | **45.36** | 10.52 | 42.98 |
| Ours | 73.44 | 71.22 | **72.77** | **56.03** | 36.82 | **72.29** | **46.10** | **46.69** | 31.13 | **84.34** | **17.26** | **71.85** | **58.41** | **32.05** | 33.13 | **23.13** | **51.67** |

eration model to produce 3D surface meshes. We evaluate the visual quality of the generated results against generative diffusion models such as NAP and CAGE, as well as objects in PartNet-Mobility.

**Metrics** For 3D segmentation, we evaluate the mean Intersection over Union (mIoU) between the predicted and ground truth point cloud segmentations. For articulation parameter estimation, we assess joint directional accuracy using angular error and joint positional accuracy using the distance between lines. For articulated object modeling, as there is no ideal metric that directly evaluate the visual quality and structural correctness of 3D objects without the need of a reference set, we instead render 2D images of articulated objects in different states and evaluate these rendered images using scores that measure the similarity between an image and a caption under the assumption that the 2D image rendered from a structurally more accurate, visually more plausible articulated object will better correspond to its category, leading to a higher image-text correspondence score. Specifically, we use CLIPscore (Hessel et al., 2021) and VQAscore (Lin et al., 2024) as our metrics for image-text correspondence.

## 4.1 QUANTITATIVE EXPERIMENTS

**3D segmentation** We compare the semantic segmentation performance of the segmentation component in ARTICULATE ANYTHING to the zero-shot version of PartSlip (Liu et al., 2023b) and the test-time alignment(TTA) version of PartDistill. The comparison used PartNetE dataset, proposed by PartSlip. Each object category in PartNetE has predefined part categories, and we adhere to this setup, evaluating segmentation performance exclusively on these predefined parts. For PartSlip and Partdistill, the input is a multi-view fusion point cloud generated by projecting RGB-D pixels from rendered images into 3D space, and the output is a semantic label for each point in the input point cloud. For ARTICULATE ANYTHING, the input is a 3D surface mesh created from a PartNetE object. It then undergoes the 3D segmentation step, producing a labeled point cloud as output. The results, shown in Table 1, demonstrate that the segmentation component of ARTICULATE ANYTHING outperforms PartSlip and Partdistill overall.

**Articulation Parameter Estimation** We define two setups for the articulation parameter estimation task:

- **Object part mesh** as input. For articulation parameter estimation on object part meshes, we compare ARTICULATE ANYTHING against two generative diffusion models, NAP and CAGE. In this experiment, the ground-truth shapes of all parts are provided, and the task is to estimate the articulation parameters for these parts. NAP is conditioned on ground-truth vertices (e.g., part bounding boxes, spatial locations, and shape latents), while the edges (joint parameters) are generated. For CAGE, certain attributes of each node, such as bounding boxes, joint types, and semantic labels, are given, while others, including joint axes and ranges, are generated. The estimated articulation parameters are compared against the ground-truth values. We evaluate this setup using articulated objects from PartNet-Mobility, following the train-test split used in CAGE. To ensure a fair comparison, we retrained NAP on the same train-test split as CAGE, as its original split differs. As shown in Table 3, our method, despite being open-vocabulary and zero-shot, achieves performance comparable to CAGE and significantly outperforms NAP within their training categories.

- **Single Observation** as input. We further compare the performance of articulation parameter estimation between ARTICULATE ANYTHING and two other methods, ANCSH and OPD. ANCSH uses a single-view point cloud as input, while OPD uses an RGB image (with optional depth information). Both methods output the segmentation of relevant parts and their joint parameters. To adapt our pipeline to this single-observation setting, we make a slight adjustment: only one image (the input observation) undergoes segmentation, and the results are directly projected as a point cloud with segmentation labels. This single-view point cloud is then processed in the second step for articulation parameter estimation.

Table 2: Quantitative evaluation of **articulation estimation**.

|  | ANCSH | OPD | Ours |
|---|---|---|---|
| error in joint direction | 6.74 | 10.37 | **5.37** |
| error in joint position | 0.065 | 0.117 | **0.049** |

Table 3: Quantitative evaluation of **Articulation Parameter Estimation**. We evaluate our estimation accuracy on PartNet-Mobility, comparing with NAP and CAGE.

| Methods | Table | Storage Furniture | Washing Machine | Safe | Dishwasher | Refrigerator | Average |
|---|---|---|---|---|---|---|---|
| Error of Joint Direction ↓ | | | | | | | |
| NAP | 13.59 | 28.66 | 29.26 | 45.28 | 53.13 | 21.37 | 31.89 |
| CAGE | **13.46** | 25.85 | **0.83** | 12.50 | **0.65** | **0.49** | 8.96 |
| Ours | 29.07 | **0.49** | 6.57 | **8.25** | 1.61 | 1.88 | **7.90** |
| Error of Joint Position ↓ | | | | | | | |
| NAP | 0.495 | 0.606 | 0.896 | 0.317 | 0.454 | 0.414 | 0.53 |
| CAGE | **0.089** | **0.174** | **0.026** | 0.126 | 0.257 | 0.144 | **0.136** |
| Ours | 0.183 | 0.559 | 0.037 | **0.116** | **0.100** | **0.143** | 0.190 |

The outcomes, as presented in Table 2, demonstrate that ARTICULATE ANYTHING outperforms ANCSH and OPD within their training domains, despite being an open-vocabulary method.

**Unconditional Generation** To evaluate our pipeline in an end-to-end fashion, we compare the quality of unconditionally generated articulated objects. Using a 3D generation model, we generate surface meshes and process them through ARTICULATE ANYTHING, extending it to an end-to-end unconditional generation pipeline. The generated articulated objects are compared to those produced by generative methods such as NAP and CAGE, as well as objects from the PartNet-Mobility dataset.

We render the test objects from multiple viewpoints and compute image-text similarity scores using the rendered images and their corresponding category names as text. The results, shown in Table 4, indicate that while the quality of textureless articulated objects generated by our pipeline is comparable to other methods, the textured counterparts exhibit higher quality compared to textured objects retrieved from PartNet-Mobility. These findings highlight that by leveraging the generative capabilities of a 3D generation model, ARTICULATE ANYTHING enables the creation of high-quality, visually appealing articulated objects.

## 4.2 QUALITATIVE EXPERIMENTS

**Unconditional Generation** We compare the appearance of objects generated by the extended pipeline of ARTICULATE ANYTHING mentioned in 4.1 with existing generative methods including NAP Lei et al. (2023), CAGE (Liu et al., 2024c) and URDFormer (Chen et al., 2024) and PartNet-Mobility dataset. All three generative methods retrieve object parts from PartNet-Mobility dataset, and that NAP and CAGE do not produce textured object parts. Therefore for fair comparison, we compare our results to NAP and CAGE in a textureless manner. The results in Figure 3 show that we achieve a comparative and even better results compared to previous works on their training categories.

Table 4: Quantitative evaluation of **Visual Quality**. We evaluate our generation quality in textured and texture-free settings, comparing with CAGE, NAP and assets retrieved from PartNet dataset.

|  | NAP | CAGE | PartNet w/o texture | Ours w/o texture | PartNet | Ours |
|---|---|---|---|---|---|---|
| CLIP Score ↑ | 0.6783 | 0.6647 | **0.6890** | 0.6563 | 0.7130 | **0.8205** |
| VQA Score ↑ | **0.6986** | 0.6178 | 0.6201 | 0.5743 | 0.6866 | **0.9376** |

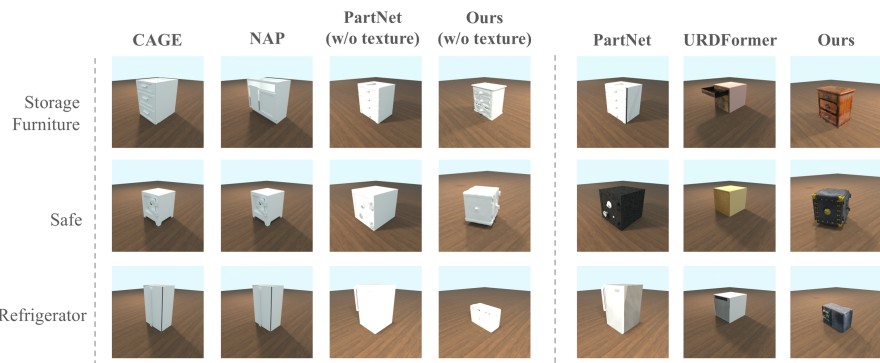

Figure 3: Visual comparison between articulated objects produced by ARTICULATE ANYTHING and other sources.

**Diversity of Modeling** Taking advantage of recent advances in generative models, the extended pipeline of ARTICULATE ANYTHING demonstrates a great diversity in aticulated object modeling. As shown in Figure 4, our extended pipeline generates high-quality articulated objects of various categories, from everyday objects to professional tools. By providing proper prompts, we can obtain customized and stylish results in the refinement step. As shown in Figure 4 (b), differentiated results can be generated from the same geometry initialization, while still preserving the key semantics of the objects. Our pipeline also features generating fictional articulated assets from anime or game series (see Figure. 4 (b)), which significantly expands the capacity of existing articulated generation methods.

**Ablation Study** We conduct ablation study on random transformation of non-fixed parts and the entire refinement step. To evaluate the effectiveness of random transformations during geometry and texture refinement, we conduct experiments where all the poses of joints are fixed. As shown in Figure 4 (e), optimizing the geometry in fixed poses can result in a loss of object part semantics (the handle is gone). Meanwhile, geometry and texture refinement contributes to an overall better visual quality of the generated asset. We ablate by comparing the geometry quality before and after the refinement step. After refinement step, finer-grained details appears on the surface, and inaccurate geometry is corrected and refined, which significantly improves the visual quality of the asset.

## 4.3 APPLICATIONS

**Annotate 3D object datasets** Other than generating from scratch, our work can also start from existing artist designed meshes. For example, we annotate part segmentation and joint structures on objects from Objaverse (Deitke et al., 2024). Such objects usually have high-quality mesh and textures and also with inner structures. Thus, we only execute the 3D segmentation part and articulation estimation part of our pipeline to annotate these objects. The results are demonstrated in Figure 5.

**Real-to-Sim-to-Real** In this experiment, we first reconstruct the 3D surface mesh of real-world objects using scanning techniques. We then apply our pipeline to convert the scanned data into an articulated object represented in URDF format, making it compatible with simulation environments. Next, we sample action targets and use motion planning (Sucan et al., 2012) to avoid collisions, generating a trajectory to successfully complete the task in simulation. Finally, we replicate the trajectory in the real world and observe that the robot arm can effectively execute the task.

We focus on three tasks involving different articulated objects: **pushing a drill trigger**, **opening a microwave door**, and **rotating a steering wheel**, as shown in Figure 6. For the pushing task, we sample pushing points and directions on the trigger's surface and generate the trajectory by applying inverse kinematics and motion planning. A trajectory rollout is considered successful if the joint position of the trigger changes in simulation. Similarly, for the **microwave door** and **steering wheel** tasks, we sample grasping poses on the surface of the handle and wheel, respectively. Pre-grasping trajectories are generated using motion planning, while post-grasping trajectories are created by heuristically rotating the end-effector along the revolute joints of the door or wheel.

Our results show that the generated trajectories can be successfully transferred to the real world, demonstrating minimal error in part segmentation and joint prediction.

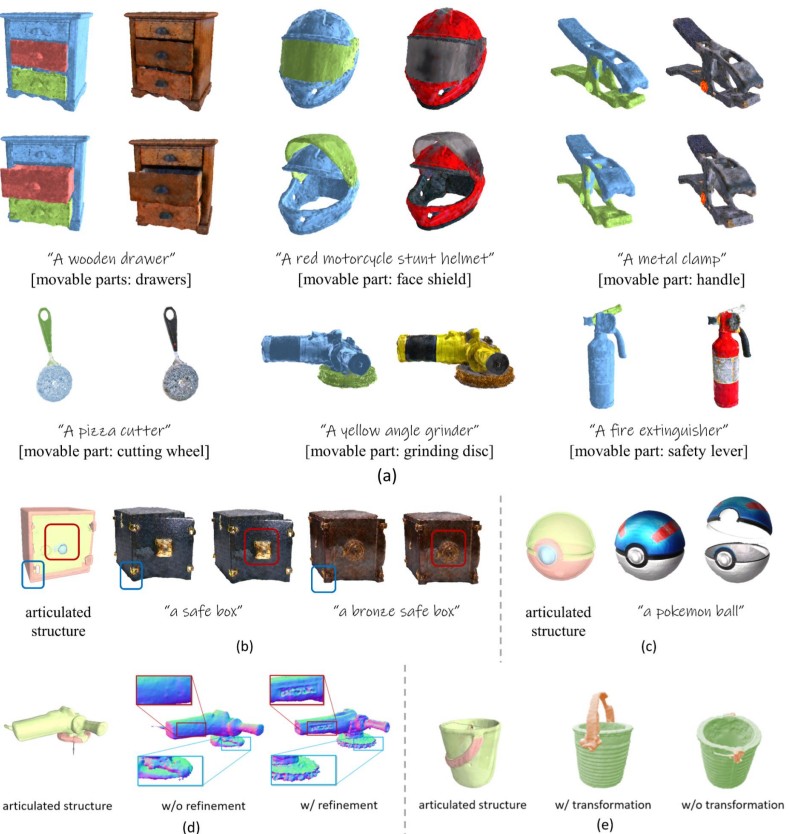

Figure 4: Visualization of articulated objects output by the refinement step. (a) Diverse articulated objects produced by ARTICULATE ANYTHING (the input meshes are generated with InstantMeshXu et al. (2024)); (b) Distinct textures created by the refinement step for the same input, with geometry and articulation parameters preserved (highlighted by the boxes); (c) A fictional object produced by ARTICULATE ANYTHING; (d) Comparison of an articulated object before and after the refinement step, showing added geometric details; (e) Comparison of refinement with and without random transformation—random transformation prevents the loss of semantic parts.

## 5   CONCLUSION

In this paper, we introduce ARTICULATE ANYTHING, a framework designed to convert open-vocabulary 3D meshes into their articulated counterparts. While previous works have effectively addressed the generation of 3D surface meshes, a key challenge remains: converting 3D meshes into articulated objects. Our pipeline first segments 3D objects based on part-level semantics, then estimates the articulation structure among object parts, and finally executes a refinement process to recover the internal structure and address issues such as holes and occlusions. By leveraging advancements in vision-language models and visual prompting techniques, we are able to segment parts and estimate articulation structure using common geometric cues. Our pipeline is capable of generating high quality textured articulated object from generated 3D meshes or ones retrieved from existing datasets. Assets we generate can support articulated object manipulation skill learning in simulation, which can be transferred to a real world robot.

**Limitations and Future work** Although our method can generate articulated objects with semantically correct parts, these articulations are not guaranteed to always be accurate and are not strictly grounded in physics. The reasons are threefold: (1) existing 3D object generation methods oftentimes generate objects with inaccurate geometry; (2) existing 3D segmentation methods lack the performance needed for reliable part-level segmentation; (3) GPT4o demonstrates bias in some situations. Addressing the above three problems to generate fully accurate and physically correct articulation parameters is an interesting avenue for future work.

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

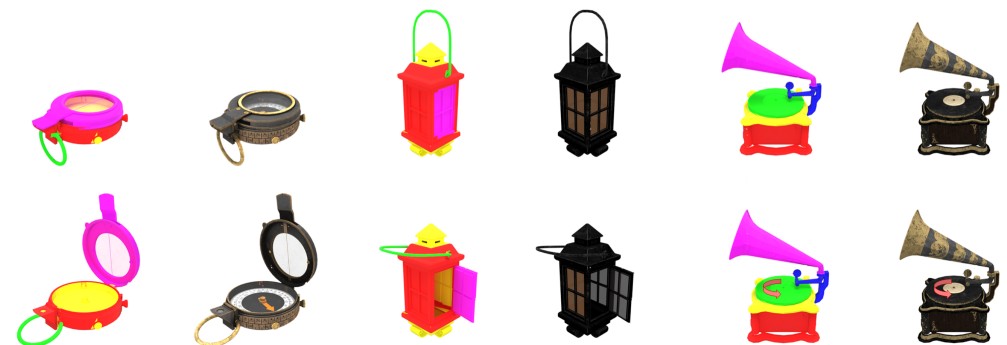

Figure 5: Our method also applies to hand-crafted 3D object. Meshes in this figure are retrieved from Objaverse

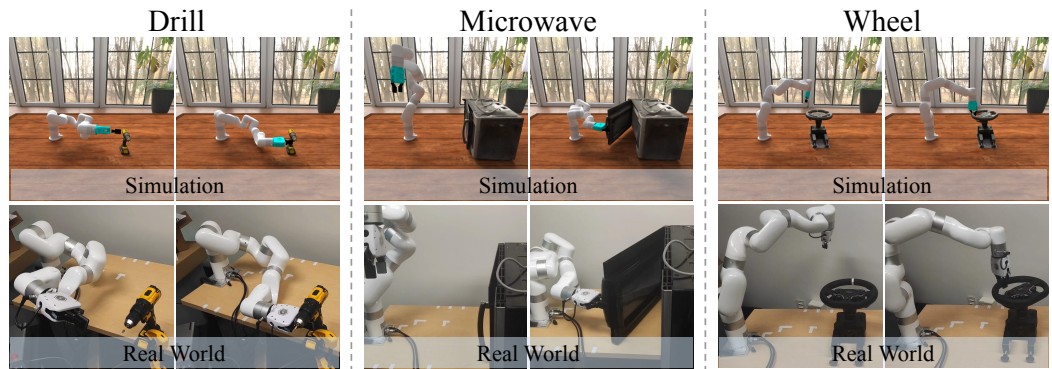

Figure 6: We build the digital twins of real-world objects in the simulation using our method. Then we sample trajectories completing the tasks in the simulation and replicate those trajectories in the real-world.

## A    REFINEMENT STAGE IMPLEMENTATION DETAILS

To implement a multi-part DMTet Shen et al. (2021) representation, we extend the vanilla DMTet backbone by assigning each object part its own independent set of point positional encoding, SDF value, and texture feature networks. During optimization, we first generate meshes through the marching tetrahedra algorithm, followed by applying random transformations to the movable parts based on forward kinematics. This approach effectively simulates the rigid transformations of articulated assets while preserving the differentiability of the rendering pipeline.

Regarding GPU memory usage, we tested our pipeline under two configurations: a standard setting using a single A100 GPU with 80GB memory, and a more constrained setting with a single V100 GPU with 32GB memory. Our pipeline consistently produced comparable results across both setups, demonstrating its efficiency in handling varying memory capacities.

## B    MORE QUALITATIVE RESULTS FROM REFINEMENT STAGE

We visualize more results generated by ARTICULATE ANYTHING generated in Figure 7

## C    DETAILS OF THE REFINEMENT STEP

Richdreamer Qiu et al. (2024) is selected as the baseline method for our optimization approach, as our method is built upon it. Richdreamer was chosen because it separately optimizes geometry and texture using two distinct diffusion models: a normal-depth diffusion model and an albedo diffusion

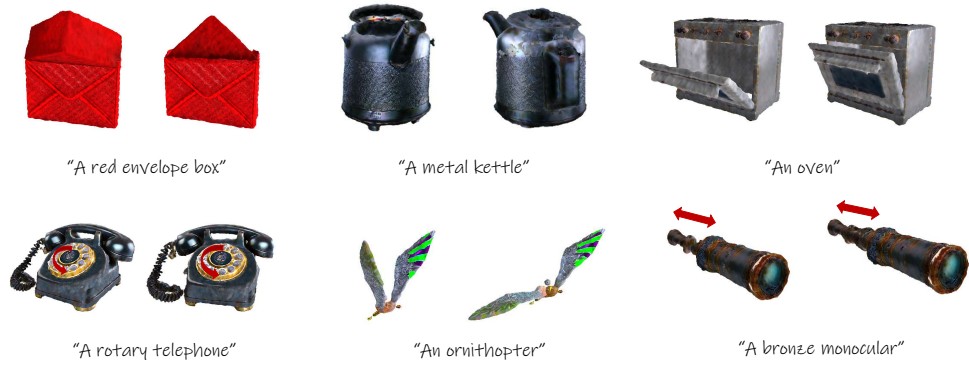

"A red envelope box"    "A metal kettle"    "An oven"

"A rotary telephone"    "An ornithopter"    "A bronze monocular"

Figure 7: Visualization of more generated assets.

model. Unlike computing the SDS loss purely in RGB space, which is suboptimal for geometry optimization, the normal-depth diffusion model demonstrates greater geometry awareness.

In Richdreamer, the Score Distillation Sampling process proposed by DreamFusionPoole et al. (2022) is formulated as follows:

Given a 3D representation $\phi$, and a differentiable renderer $g$, the rendered image is $x = g(\phi)$. The SDS loss is then used to optimize the 3D representation $\phi$:

$$\nabla_\phi \mathcal{L}_{\text{SDS}}(\phi, x = g(\phi)) = \mathbb{E}_{t,\epsilon}\left[w(t)(\epsilon_\theta(z_t; y, t) - \epsilon)\frac{\partial x}{\partial \phi}\right],$$

where $z_t$ is the noisy latent code, $\epsilon$ is the injected noise and $\epsilon_\theta$ is the noise predicted by a denoising model $\theta$, conditioned on timestep $t$ and text embedding $y$. The term $w(t)$ is a timestep dependent weighting factor.

In Richdreamer and other previous works the 3D representation $\phi$ is static, whereas in our case, it is articulated. Omitting other attributes, we denote the 3D representation of articulated objects as $\phi_q$, where $q$ is a vector representing joint positions. During optimization, the base of the articulated object remains fixed. Since non-fixed joints of interest (revolute, prismatic, and continuous) all have one degree of freedom, each element in $q$ corresponds to the position of a non-fixed joint in $\phi_q$. The articulated object in its rest configuration is denoted as $\phi_{q_0}$. A transformation function $T$ maps $\phi_{q_0}$ to $\phi_q = T(\phi_{q_0}, q)$ given the desired joint positions $q$.

The optimization process for Richdreamer and other previous SDS optimization methods can be briefly summarized by the following pseudo-code:

```
FOR i IN iterations:
    render an image x (x = g(phi))
    sample timestep t
    compute SDS loss
    update optimizer
```

In our approach, we extend this process by sampling joint positions and transforming object parts accordingly:

```
FOR i IN iterations:
    sample joint position q
    transform parts according to q (phi_q = T(phi_q0, q))
    render an image x (x = g(phi_q))
    sample timestep t
    compute SDS loss
    update optimizer
```

Table 5: Quantitative ablation study of refinement step.

|  | No Refinement | Refinement w/o transformation | Refinement w/ transformation |
|---|---|---|---|
| CLIPScore | 0.7329 | 0.7928 | **0.8205** |
| VQAScore | 0.6551 | 0.8164 | **0.9376** |

## D  EXPERIMENT SETTINGS

**articulation parameter estimation**  The quantitative result for articulation parameter estimation are presented in Table 3. We use objects from CAGE's test split. In our setup, the shapes of the testing objects are known, and articulation parameters are predicted. For NAP, ground truth vertices (e.g., part bounding boxes, spatial locations, shape latents) are provided, while edges (joint parameters) are estimated. Since NAP uniformly represents all joints using Plücker coordinates, we evaluate only the translational component for prismatic joints and the rotational component for revolute joints. For CAGE, some attributes of each node, such as bounding boxes, joint types, and semantic labels, are provided, while others, including joint axes and ranges, are estimated. In AR-TICULATE ANYTHING, the shapes, semantics of each part, and joint types are given, and the joint axes and limits are estimated. We evaluate the estimated joint axes against the ground truth using angle error and line distance, as described in Section 4, Metrics.

**unconditional generation**  The quantitative results for the visual quality of unconditionally generated articulated objects are presented in Table 4. We generate objects using minimal input. For NAP, no conditions are provided; its diffusion model generates objects unconditionally. After generation, the initial shape is decoded from the generated shape latent and replaced by the nearest matching part mesh. For CAGE, a random articulated object from PartNet-Mobility, in CAGE's test split, is retrieved. Its connectivity graph and object category label are used as input to CAGE's diffusion model. After generating bounding boxes, part semantics, and joint parameters, part meshes are retrieved using CAGE's retrieval method. For URDFormer, an object category is randomly sampled, and an image is generated using Stable Diffusion 3 (prompted to produce a front-view image of an object in the sampled category with a white background). URDFormer then takes the generated image as input and outputs a URDF. In ARTICULATE ANYTHING, an image is generated similarly, followed by mesh generation using InstantMesh. The generated mesh is processed through the full pipeline of ARTICULATE ANYTHING to produce an articulated object. For PartNet-Mobility, objects in the relevant categories are randomly retrieved.

For each generated object, we render eight views surrounding it and compute the CLIPScore and VQAScore. The input text for these scores is "a OBJECT_CATEGORY". Figure 3 is the visualization of some unconditionally generated articulated objects.

## E  ADDITIONAL EXPERIMENTS

**More Ablation Study of Refinement Step**  In the ablation study, we evaluate the same objects used in the quantitative evaluation of unconditional generation. We apply three different settings to the intermediate output of the articulation estimation step:

No refinement step applied. Refinement step applied without random transformation (same as Rich-dreamer). Refinement step applied with random transformation. The results are presented in Table 5. The highest scores are achieved when using refinement with random transformation, demonstrating the effectiveness of our refinement step.

We also provide visualizations of the input and output of the refinement step, as shown in Figure 8. The pencil case and lighter in the second and third rows feature manually set joint parameters and shape primitives as link geometries. The refinement step successfully optimizes storage space for the drawer and pencil case and generates a nozzle structure for the lighter, and produces plausible textures.

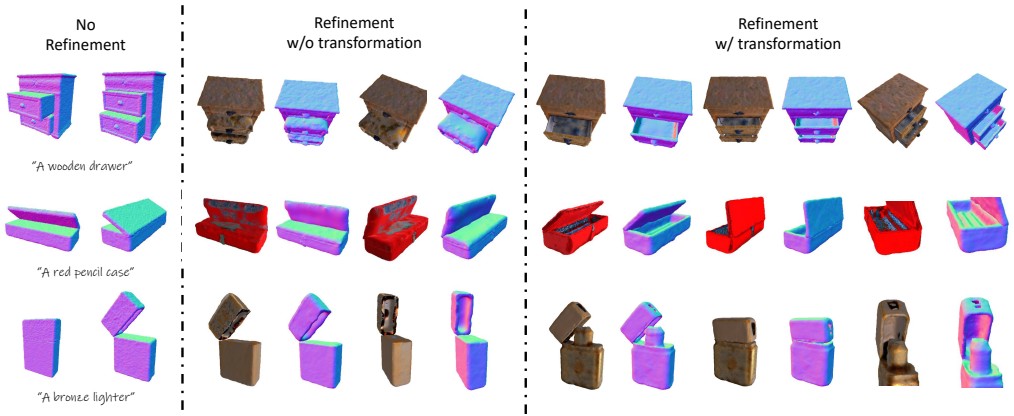

Figure 8: Visualization of the input and output of the refinement step.

## F PROMPTING DETAILS

**3D segmentation** The 3D segmentation process involves the following steps:

1. Render the mesh to obtain multi-view images.

2. Apply SAM to generate segmentation masks for each rendered image.

3. Label the generated masks on the images using techniques proposed by SoM.

4. Provide the unlabeled images to GPT4o to identify object parts using the following prompt:

```
You have a good understanding of the structure of articulated objects. Your job is to assist the
user to analyze the structure of an object. Specifically, the user will give you an image of an
articulated object, and your task is to recognize the main parts of that object. You should give
your answer in the following format:

```part_list
(1) part_name: name of the part; description: a brief description about the part, and how it moves
(2) part_name: name of the part; description: a brief description about the part, and how it moves
...

```

Remember:
(1) Do not answer anything not asked.
(2) Your answer should be purely based on the input image, do not imagine anything.
(3) If there are multiple parts with the same semantic, just add one part to the list. For
example, if there are four wheels, just add one part whose name is wheel.
```

5. Prompt GPT4o to generate the link and joint configuration of the articulated object, and exclude non-movable parts, using the following prompt:

```
SYSTEM PROMPT:

You have a good understanding of the structure of articulated objects. You are very familiar with
URDF format. Your job is to assist the user to analyze the structure of an articulated object.
Specifically, the user will name an object and then give you the main parts of that object. You
will have to group these parts into links and then give the joints connecting these links. You
should give your answer in the following format:

```articulation tree
parts:
(1) part_name: name of the recognized part;
...

links:
(1) link_name: name of the link;
...

joints:
(1) joint_name: name of the joint; joint_type: type of the joint; parent_link: name of the parent
link; child_link: name of the child link; joint_limit: [lower limit, upper limit];
...
```

For example:
```

```
```articulation tree
parts:
(1) part_name: Front windshield;
(2) part_name: Doors;
(3) part_name: Headlights;
(4) part_name: Wheels;
(5) part_name: Windows;

links:
(1) link_name: Chasis;
(2) link_name: Doors;
(3) link_name: Wheels;
(4) link_name: Windows;

joints:
(1) joint_name: wheel_chasis_joint; joint_type: continuous; parent_link: Chasis ; child_link:
Wheels; joint_limit: None;
(2) joint_name: door_chasis_joint; joint_type: revolute; parent_link: Chasis ; child_link: Doors;
joint_limit: [0, 90];
(3) joint_name: door_chasis_joint; joint_type: prismatic; parent_link: Chasis ; child_link:
Windows; joint_limit: [0, 1];

```

Remember:
(1) Do not answer anything not asked.
(2) If a part is actually movable in any direction, its joint type is floating.
(3) Available joint types are: fixed, prismatic, revolute, continuous and floating.
(4) For joint_type, only answer one word (among the available types), do not answer anything else.
(5) For every part that is not fixed, there must be a unique link for it.
(6) For parts that are fixed, try to group as many as you can.
(7) Joint limit must be given for prismatic joints and revolute joints. The unit of a revolute
joint limit is degrees. The unit of a prismatic joint limit is the size of its child link in its
translation direction. The joint limit should be two numbers.

USER PROMPT:

Object: OBJECT_NAME
Parts: RECOGNIZED_PARTS
```

6. Provide the labeled images to GPT4o and request it to group the segmentations based on their semantics:

```
The image is of a {object_name}. I have labeled a bright numeric ID at the center for each visual
segment of the object in the image. Please group these segments into semantic parts. Give your
answer in the following format:

```part_list
1. name: name of the part; labels: the ID of segments that belong to this part
2. name: name of the part; labels: the ID of segments that belong to this part
...
```

The parts we care about are:
LINK_NAMES

Remember:
(1) Sometimes multiple segments belong to the same part.
(2) We only care about the listed parts, do not add anything else to the part list.
(3) You do not necessarily have to assign every segment to a part, you may omit some of them if
they belong to none of the parts we care about.
(4) If some of the listed parts do not exist in the image, omit these parts.
(5) If there are multiple instances of a semantic part in the image, add two instances of that
semantic part to the part list. For example, if a fridge has two doors, add instance door@1 and
door@2 (the format is instance_name@unique_id).
```

7. Merge masks based on their overlap ratio, following the method described in Part123 Liu et al. (2024a). The semantics of each merged mask is determined by the most frequent semantic label among its 2D mask components. A merged mask is represented as a 3D point cloud, obtained by concatenating the projections of its 2D mask components into 3D space.

8. Further merge the masks based on connectivity. Two adjacent 3D masks $M_1, M_2$ with the semantics label are merged unless there exists at least one pair of 2D masks $m_1, m_2$, where $m_1$ is a 2D mask component of $M_1$, $m_2$ is a 2D mask component of $M_2$, such that $m_1$ and $m_2$ are two different instances of the same semantic part, and they co-occur in at least one image.

**articulation parameter estimation**   After the 3D segmentation step, we obtain a set of 3D semantic masks represented as pointcloud $\{M_1, M_2, ..., M_n\}$, along with the overall pointcloud of the surface mesh $M_{all} = M_1 \cup M_2 \cup ... \cup M_n$. The articulation parameter estimation involves the following steps:

1. Identify the connected area. For the part currently under consideration, denote its pointcloud as $M_i$ and the remaining pointcloud as $M_{rest} = M_{all} \setminus M_i$. The mesh corresponding to $M_i$ is cropped, and its SDF is computed. For each point in $M_{rest}$, we compute its SDF value. Points in $M_{rest}$ with an SDF value below a predefined threshold are identified as the connected area.

2. Identify a set of key points. We then apply DBSCAN to cluster the connected area. Then each cluster is further divided by KMEANS. The cluster centers are considered as the key points. We then project these points onto the rendered 2D images.

3. Prompt GPT4o to extract the articulation parameters. The prompt varies depending on the joint type.

For revolute joints, the prompt first determines whether both points of the joint are on the surface:

```
You are an assistant with a deep understanding of the structure of objects. Your task is to help
users determine the hinge position of some parts of a given object mesh using common sense. It is
important to note that the object is represented by a mesh, so you only have access to the
object's surface and no access to its inner structure. The term "hinge" here does not only refer
to the mechanical structure of a hinge, but also has a broader meaning. For example, the
connection between a cardboard box lid and the body of the box is also considered a hinge.

Specifically, the user will provide you with an object, and the part for which the hinge position
needs to be predicted will be specified. You will have to decide whether (1) both ends of the
hinge are positioned on the surface of the object or (2) only one end of the hinge is positioned
near the surface of the object, and the other end is inside the object.

For example, the hinge of a door has both its ends positioned on the door frame, which is
recognizable and falls into the first category. The hinge of a wheel has one end recognizable on
the center of the wheel, and its other end hidden inside the car (normally an object mesh of a car
will not have detailed mechanical structures), which falls into the second category.

Please give your answer in the following format:
```hinge_info
description: a brief description of the hinge
choice: (1) or (2)
```
```

Based on GPT4o's selection, we provide a tailored prompt. For choice (1), the specific prompt is as follows. Once GPT4o selects two or more points, we fit a line through these points, defining the rotation axis.

```
You are an assistant with a deep understanding of the structure of objects. Your task is to answer
some questions about the input image of an object. The input image is of a {object_name}. The
image has some points marked, each with an numerical ID as a label. Please select the points that
are on the rotation axis of the {part_name} of the {object_name}. Give your answer in the
following format:

```hinge points
description: a brief description
selected IDs: ID of selected point1, ID of selected point2, ... (for example 1,3)
```

Remember:
(1) Do not answer anything not asked for.
(2) Select two or more points.
(3) Give your answer based on the provided image.
```

For choice (2), the prompt is as follows. The rotation axis is determined using the point selected by GPT4o and the normal of the connected area.

```
You are an assistant with a deep understanding of the structure of objects. Your task is to answer
some questions about the input image of an object. The input image is of a {object_name}. The
image has some points marked, each with an numerical ID as a label. Please select the point that
is on the rotation axis of the {part_name} of the {object_name}. Give your answer in the following
format:

```hinge points
description: a brief description
selected IDs: ID of the selected point
```

Remember:
(1) Do not answer anything not asked for.
(2) Only select one point that is the most suitable.
(3) Give your answer based on the provided image.
```

For prismatic joints, we prompt GPT4o to determine whether its child link moves in and out or along the surface:

```
You are an assistant with a deep understanding of the structure of objects. Your task is to help
users to determine the translation direction of some parts of a given object mesh using common
sense. It is important to note that the object is represented by a mesh, so you only have access
to the object's surface and no access to its inner structure.
```

```
Specifically, the user will provide you with an object and the part for which the translation
direction needs to be predicted will be specified. You will have to decide whether the translation
direction is outwards from/inwards towards the mesh, or along the surface of the mesh.

When a part moves outwards, you will see more of that part coming out from the object. When a part
moves inwards, you will see portions of that part going into the object. When a part moves along
the surface of an object, you still see the exact same part.

For example, a pressing button can be pressed inwards (when you press it, the button goes into the
object), the telescopic handle of a suitcase can be pulled outwards (when you pull it, the entire
handle comes out of the suitcase), and a stick shift moves along the surface of a shift pattern
(when you are shifting, the shift does not go into the transmission or out of the transmission).

Please give your answer in the following format:
```translation_axis_info
description: a brief description of the translation axis
choice: outward/inward or surface
```
```

If the child link moves inward or outward, the translation direction is defined by the normal vector of the connected area. If it moves along the surface, we draw four arrows (originating from the child link's center and pointing up, down, left, and right) on the image and prompt GPT4o to select the translation direction:

```
You are an assistant with a deep understanding of the structure of objects. Your task is to answer
some questions about the input image of an object. The input image is of a {object_name}. The
image has some arrows marked, each with a different color. Please select the arrow that indicate
the translation direction of the {part_name} of the {object_name}. Give your answer in the
following format:

```hinge points
description: a brief description
selected arrow: color of the arrow (in red, yellow, blue, green)
```

Remember:
(1) Do not answer anything not asked for.
(2) Select one arrow.
```

The selected arrow is then projected onto the plane fitted to the connected area, and the translation direction is defined by the direction of the projected arrow.

4. Validate joint limits. For revolute joints, we incrementally rotate the child link based on the joint parameters and identify the maximum range where penetration remains below a predefined threshold. For prismatic joints, we follow the joint limits provided by GPT4o.

