# OpenReview forum: "Articulate Anything: Open-vocabulary 3D Articulated Object Generation"
_ICLR.cc/2025/Conference — Submitted to ICLR 2025_

### Official Review · Reviewer_E2pY · 2024-10-23

**Soundness:** 2
**Presentation:** 2
**Contribution:** 3
**Rating:** 5
**Confidence:** 4

**Summary:**

This paper works on the task of converting a 3D mesh into its articulated counterpart, and there are two scenarios:
1) if the input mesh comes with part geometries, this work can segment the articulated parts and estimate the articulation parameters for each part so that the mesh can be articulated;
2) if the input mesh is a single surface, this work takes an additional refinement step to generate an articulated object which takes the input mesh as initialization and a text as additional input.

This work proposes a three-stage pipeline to work with arbitrary input, which is stated as open-vocabulary in the paper.
- In the first stage, it proposes to leverage VLM to segment the part in 3D from multi-view images rendered from the input mesh.
- In the second stage, it first proposes several candidate 3D points where the joint might appear based on several heuristic rules. Then the VLM is prompted to select the points on the image to infer the joint parameters in 3D.
- In the third stage, it first reconstructs DMTet for each part and then uses SDS loss to refine the incomplete regions.

The main contribution is this pipeline that enables the generation of diverse articulated objects by taking arbitrary 3D mesh as input.

**Strengths:**

- This paper identifies a critical research gap in 3D generation for articulated objects and contributes to an increasingly important area.
- This paper proposes a novel pipeline that enables the creation of articulated objects from arbitrary input mesh.
- The paper shows promising preliminary results to demonstrate the effectiveness of the method.

**Weaknesses:**

- **Somewhat misleading presentation**: I believe the paper would benefit from reorganization to better clarify the main goal or task that this work addresses, and the downstream applications that the proposed method enables. The way the paper currently presents its goal and teaser figure seems somewhat misleading. Based on my understanding, there are essential two tasks enabled by the pipeline proposed in the paper. The high-quality output shown in Figure 1 is produced using an input mesh that already contains part geometries, which aligns with the stated goal of "converting a rigid mesh into its articulated counterpart” as the first task.
The second task involves input meshes without any part geometries, requiring an additional reconstruction and optimization step. Based on my understanding, this process cannot fully preserve the original input in terms of geometry and appearance. This feels more like a text-to-3D generation task, where a 3D mesh serves as an initialization and a text input provides conditional guidance, rather than a straightforward conversion.
- **No quantitative evaluation of the reconstruction quality**: The paper lacks a quantitative evaluation of the reconstruction accuracy, in terms of geometry and appearance. I believe this is one of the most critical experiments needed to validate the approach. Specifically, taking a mesh surface (without part geometry) of the object in PartNet-Mobility as input, once it is reconstructed using the proposed pipeline, the Chamfer Distance of the part meshes with respect to the ground truth object can be reported. To consider the different states of the articulated objects, the ID and AID metrics (proposed by NAP and CAGE) can also be reported. For appearance, the PNSR/SSIM/LPIPS can be reported by rendering multi-view images.
- **Missing Technical and Experimental Details**: While the overall idea of the paper is easy to follow, many critical technical and experimental details are omitted. This lack of detail significantly weakens the paper’s reproducibility. Also,  it is unclear whether the comparison experiments presented are conducted in a fair manner. Please refer to the `question` section for specific requests for clarification.
- **Low resolution of Qualitative Results**: The resolution of most qualitative results, particularly those in Figures 3, 4, 5, and 6, is relatively low. I strongly recommend replacing these with higher-resolution images that better showcase the output quality. Based on the current results, it seems to me that the reconstructed objects shown in Figure 8 are of much higher geometry quality compared to other results. This inconsistency raises concerns about the overall quality of the generated outputs. I am not fully convinced by the quality of the generated objects as currently presented.

**Questions:**

There are **several details that need clarification**. Providing these details would help clarify the evaluation process and ensure the results are reproducible.
- How is the GPT4o exactly prompted for the task of part segmentation and joint point selection?
Once the joint point is selected, how are the joint limit and the direction of rotation/translation determined?
- How is the SDF of each part computed in section 3.3?
- How is the optimization process implemented? Where is the text prompt from? How many iterations are required to optimize each object? How long does it take?
- For unconditional generation, how is the experiment conducted? What are the input to this method and other baselines (NAP, CAGE, URDFormer)? Did you retrain URDFormer on the same split as CAGE?
- What do the red and blue boxes mean in Figure 5? Is the object shown at the left-most the input?
- For the comparison of articulation parameter estimation, what is the input to NAP and CAGE? How are they implemented to be compared?

**Other questions about the experiment results**:
- In Table 2, the "Ours" score in the last column is significantly higher than that of PartNet-Mobility, which I assume serves as a ground-truth reference. Could you clarify why this is the case?
- In Table 2, the VQA score for NAP is also higher than “PartNet w/o texture”. What is it implied? It would be helpful to understand the reasoning behind this discrepancy and how the scores are being interpreted.
- In Figure 3, what is the input to each method and how are the examples selected for comparison?


**Missing comparison points**:

On the side of articulation estimation, it is possible to compare it with other methods beyond just NAP and CAGE, such as Shape2Motion and Real2Code.

**Missing references**:

[1] Hu, Ruizhen, et al. "Learning to predict part mobility from a single static snapshot." ACM Transactions On Graphics (TOG) 36.6 (2017): 1-13.

[2] Sharf, Andrei, et al. "Mobility‐trees for indoor scenes manipulation." Computer Graphics Forum. Vol. 33. No. 1. 2014.

[3] Weng, Yijia, et al. "Neural Implicit Representation for Building Digital Twins of Unknown Articulated Objects." Proceedings of the IEEE/CVF Conference on Computer Vision and Pattern Recognition. 2024.

[4] Wei, Fangyin, et al. "Self-supervised neural articulated shape and appearance models." Proceedings of the IEEE/CVF Conference on Computer Vision and Pattern Recognition. 2022.

[5] Liu, Jiayi, Manolis Savva, and Ali Mahdavi-Amiri. "Survey on Modeling of Articulated Objects." arXiv preprint arXiv:2403.14937 (2024).

---

> ### Author Response · Authors · 2024-11-25
> **Response to Reviewer E2pY [1/3]**
>
> Thank you for your insightful and constructive comments! We have added additional experiments and modified our paper according to your comments.
>
> ### 1. Somewhat misleading presentation
> Thank you for pointing out the inconsistency between the task proposed in our paper and its presentation. We acknowledge that the primary goal of Articulate Anything is to convert a rigid mesh into its articulated counterpart. The refinement step is an additional process designed to make our pipeline complete when the input mesh is a surface mesh. We believe this refinement step aligns with the goal of converting a rigid mesh into its articulated counterpart because, although it modifies the geometry and texture of the input surface mesh, it preserves the semantic parts and articulation parameters. Since our ultimate goal is to collect large-scale, realistic articulated object data for robotics and embodied AI, preserving the geometry and texture is not essential, particularly when they are low-quality or unrealistic.
>
> Additionally, open-vocabulary articulated object generation represents a novel downstream task enabled by our pipeline. This is achieved by leveraging a 3D generation model to create 3D surface meshes as inputs for Articulate Anything. The unconditional experiments were conducted to evaluate the end-to-end performance of our pipeline.
>
> With these clarifications, we have reformed our paper to make it more self-contained and aligned with its core objectives.
>
> ### 2. No quantitative evaluation of the reconstruction quality
>
> As discussed earlier, our goal is to collect large-scale, realistic articulated object data for robotics and embodied AI. Therefore, we prioritize preserving articulation parameters and part semantics over the low-quality geometry and texture of generated surface meshes. Guided by this perspective, we evaluate the refinement step using metrics that assess the visual quality and realism of refined objects, rather than reconstruction accuracy.
>
> While objects in PartNet-Mobility can serve as ground truth for articulation parameters and part semantics, they are synthetic in terms of rendering quality, with simplistic textures, materials, and shapes. Additionally, the diversity of objects in PartNet-Mobility is limited. As a result, we believe that PartNet-Mobility objects cannot be used as ground truth for geometry and texture. The goal of our pipeline is not to fit the distribution of PartNet-Mobility but to go beyond existing datasets and collect realistic articulated objects.
>
> Therefore, we use image-based scores rather than metrics like ID, which measure the distance between our collected objects and those from PartNet-Mobility. For articulation parameters and part semantics, we conducted additional experiments to compare articulation parameter estimation performance with methods focused on this task, such as ANCSH and OPD. The results are detailed in General Response Section 3B.
>
> ### 3. Missing Technical and Experimental Details
> To enhance clarity and reproducibility, we have expanded our appendix with the following sections:
>
> - **Section C: Details of the Refinement Step**: This section provides a comprehensive description of the refinement step.
> - **Section D: Experiment Settings**: Here, we clarify the experimental setups for articulation parameter estimation and unconditional generation, including the inputs and outputs for both baseline methods and our method, which ensures fair comparison.
> - **Section F: Prompting Details**: This section includes the exact prompts used for GPT-4 and outlines the detailed substeps for the 3D segmentation and articulation parameter estimation steps.
>
> ### 4. Low resolution of Qualitative Results
> Thank you for the reminder. We have updated the figures with high-resolution versions to improve clarity and presentation quality.

---

> ### Author Response · Authors · 2024-11-25
> **Response to Reviewer E2pY [2/3]**
>
> ##### Questions:
>
> > **Q1: How is the GPT4o exactly prompted for the task of part segmentation and joint point selection?**
>
> Please refer to Appendix F: Prompting Details.
>
> > **Q2: Once the joint point is selected, how are the joint limit and the direction of rotation/translation determined?**
>
> A rotation axis has six unknowns and requires at least two points in 3D space or one point plus a directional vector to define it deterministically. (If more than two points are available, a line is fitted through them.) A translation axis, having three unknowns, requires only a directional vector to define it.
>
> For revolute joints, we incrementally rotate the child link based on the joint parameters and identify the maximum range where penetration remains below a predefined threshold.
> For prismatic joints, we follow the joint limits provided by GPT4o.
>
> For details on determining the joint axis, refer to Appendix F: Prompting Details, paragraph Articulation Parameter Estimation, substep 3. For determining the joint limit, refer to substep 4 of the same paragraph.
>
> > **Q3: How is the SDF of each part computed in section 3.3?**
>
> The 3D segmentation step generates a segmented point cloud for each part. For each part, the Signed Distance Function (SDF) of the mesh corresponding to its segmented 3D point cloud is computed. Then, for points outside the part, their SDF values are evaluated. Points with SDF values below a predefined threshold are identified as the connected area.
>
> For further detail, please refer to Appendix F: Prompting Details, paragraph Articulation Parameter Estimation, substep 1.
>
> > **Q4: How is the optimization process implemented? Where is the text prompt from? How many iterations are required to optimize each object? How long does it take?**
>
> We have included the detailed implementation of the optimization process in General Response Section 4A.
>
> In the quantitative experiments, the text prompt used is simply "a OBJECT_CATEGORY_NAME," while the text prompts for qualitative results (Figures 4, 5, and 9) are manually specified. Geometry refinement requires 1,000 iterations, and texture refinement takes 1,600 iterations. Using a single A100 GPU with a rendering resolution of 1024x1024, geometry refinement takes approximately 50 minutes, and texture refinement takes 90 minutes.
>
> > **Q5: For unconditional generation, how is the experiment conducted? What are the input to this method and other baselines (NAP, CAGE, URDFormer)? Did you retrain URDFormer on the same split as CAGE?**
>
> NAP takes no input. CAGE requires a connectivity graph and an object category label as input. URDFormer takes an image as input. Articulate Anything takes a text input, specifically the object category name. For more details of the experimental setup of unconditional generation, including the inputs for Articulate Anything and other baseline methods, please refer to General Response section 4B.
>
> As URDFormer has not released their training code, we used the checkpoint they provided.
>
> > **Q6: What do the red and blue boxes mean in Figure 5? Is the object shown at the left-most the input?**
>
> The boxes are intended to demonstrate that the semantic features of parts are preserved across different text prompts during the refinement stage. The object shown at the left-most is a visualization of part segmentation, not the input.
>
>
> > **Q7: For the comparison of articulation parameter estimation, what is the input to NAP and CAGE? How are they implemented to be compared?**
>
> For NAP, ground truth vertices (e.g., part bounding boxes, spatial locations, shape latents) are provided, while edges (joint parameters) are estimated. For CAGE, some attributes of each node, such as bounding boxes, joint types, and semantic labels, are provided, while others, including joint axes and ranges, are estimated. We use the official implementations of NAP and CAGE from their respective GitHub repositories.
> More details about the comparison of articulation parameter estimation is described in General Response section 3B.

---

> ### Author Response · Authors · 2024-11-25
> **Response to Reviewer E2pY [3/3]**
>
> > **Q8: In Table 2, the "Ours" score in the last column is significantly higher than that of PartNet-Mobility, which I assume serves as a ground-truth reference. Could you clarify why this is the case?**
>
>
> As previously discussed, objects in PartNet-Mobility serve as ground truth for kinematic structure. However, in terms of rendering quality, they are synthetic and feature simple textures and materials. In contrast, the 2D diffusion model used during optimization is trained on rendered views of Objaverse objects, which exhibit higher visual quality compared to PartNet-Mobility objects. This allows the optimized texture to appear more visually plausible than that of PartNet-Mobility objects.
>
> Furthermore, Richdreamer optimizes a Physically Based Rendering (PBR) material model, further enhancing visual quality. For instance, the safe in Figure 5 has a metallic surface, while the drawer in Figure 4 appears wooden. These factors collectively contribute to the higher scores achieved. However, when comparing textureless renderings (columns 3 and 4 in Table 2), the score of Articulate Anything no longer surpasses that of PartNet-Mobility.
>
> > **Q9: In Table 2, the VQA score for NAP is also higher than “PartNet w/o texture”. What is it implied?**
>
> NAP retrieves part meshes from PartNet-Mobility and only slightly exceeds PartNet w/o texture in performance. We believe this small difference is likely due to random fluctuations.
>
> > **Q10: In Figure 3, what is the input to each method and how are the examples selected for comparison?**
>
> The input to each method (NAP, CAGE, URDFormer) is the same as in the unconditional generation setting. The examples are selected randomly.
>
> > **Q11: On the side of articulation estimation, it is possible to compare it with other methods beyond just NAP and CAGE, such as Shape2Motion and Real2Code.**
>
>
> We have added comparisons with ANCSH and OPD for articulation estimation, with the results presented in General Response Section 2C. Since Shape2Motion does not provide a checkpoint trained on PartNet-Mobility, we selected two more recent works, ANCSH and OPD, which offer trained checkpoints. As for Real2Code, their finetuned LLM checkpoint and inference code have not been released, so we cannot make a comparison.
>
> > **Q12: Missing references**
>
> Thanks for the reminder. We have incorporated the missing citations into our paper.
>
> *We wish that our response has addressed your concerns, and turns your assessment to the positive side. If you have any more questions, please feel free to let us know during the rebuttal window.*
>
> Best,
>
> Authors

---

> > ### Comment · Reviewer_E2pY · 2024-11-28
> >
> > Thank the authors for the detailed responses and the huge effort made in the revision.
> > The additional experiments and explanations make the paper more comprehensive, which solves some of my concerns.
> >
> > However, the current revision seems to represent a significant shift in focus from generation to perception, which alters the paper’s core contribution. In that case, I think it might require a longer period to revise the story and restructure the experiments to put together a coherent paper. At present, the assumptions, contributions, and applications are not clearly articulated. Since the validity of the experiments depends on the clarity of these claims, it is difficult to assess whether the current experimental setup and evaluation adequately support the proposed method.
> >
> > So I decided to keep my original score.

---

> > > ### Author Response · Authors · 2024-11-29
> > >
> > > hank you for your thoughtful feedback and for recognizing the effort we made in revising the paper. We appreciate your acknowledgment that the additional experiments and explanations have addressed some of your concerns.
> > >
> > > We would like to clarify a few key points regarding your feedback:
> > >
> > > 1. **Core Contribution and Focus**
> > >    The primary contribution of our paper has always been the proposed pipeline. The generation capability, while important, is presented as a crucial application and a secondary contribution enabled by the proposed pipeline. This focus has been consistent throughout the revisions.
> > >
> > > 2. **Scope of Revisions**
> > >    While we understand your observation regarding a perceived shift in focus, we want to emphasize that the changes in our revision were primarily confined to the experimental section. These updates were aimed at providing additional evaluations of the proposed pipeline to reinforce its effectivenes, not to alter the paper’s narrative or contributions. The introduction and methodology sections remain largely unchanged, reflecting our original intentions.
> > >
> > > 3. **Experimental Setup and Validity**
> > >    If there are specific aspects that remain unclear, we would be happy to provide further clarification or additional supplementary materials to ensure our claims are fully supported.  Regarding the experimental setup, our revisions were designed to demonstrate the robustness of our method against prior works and validate its performance. If you believe there are additional perspectives or metrics that could strengthen the evaluation, we would greatly appreciate your suggestions.
> > >
> > > We sincerely value your feedback and are committed to improving the clarity and coherence of our work. We hope this explanation addresses your concerns, and we remain open to further discussion.

---

### Official Review · Reviewer_z2rV · 2024-11-03

**Soundness:** 3
**Presentation:** 3
**Contribution:** 3
**Rating:** 6
**Confidence:** 4

**Summary:**

In this paper, the authors propose an automated framework, that converts rigid 3D surface meshes into articulated meshes. It first uses VLM to segment the object into parts, then uses geometric cluses and visual prompting to estimate joint parameters. Finally it refines the parts through SDS optimization. It improves existing optimization method by randomly transform the parts during the optimization process.

The method can be applied on AIGC（Artificial Intelligence Generated Content） mehes and hand-crafted 3D models. Experimental results show that it can generate high-quality meshes. Experiments on PartNet-Mobility show that it can estimate the joint parameters accurately.

**Strengths:**

This paper designs an effective pipeline for open-vocabulary 3D articulated object generation. It leverages the advanced VLM and visual promptiing techniques to segment parts and estimate the joint parameters.

The integration of all three modules to achieve high performance and successfully connect them is impressive. To the best of my knowledge, this is the first work on open-vocabulary 3D articulated object generation.

The application of the proposed method on Real-to-Sim-to-Real is interesting.

**Weaknesses:**

My main concern is the Part C （Geometry & Texture Refinement）.

The manuscript in Part C is too rough. To my understanding, it is an improvement to existing SDS optimization method Qiu et al. (2024). If so, the baseline method should first be desribed, making the paper self-contained. Further detailed description of the improvement is lacking. The authors should add some figures to illustate the optimization pipeline and some math equations should be added.

As claimed, the refinement process should be able to generate and optimize the inner structure of each part, but it is difficult to see and evaluate the performance based on the figures in the submission or the webpage.
I suggest providing cross-sectional views of refined objects, including before-and-after comparisons of internal geometries,quantitative metrics to evaluate the quality of inner structures

Provide examples of cases where the pipeline fails or produces suboptimal results.
Analyze the root causes of these failures.
Discuss potential solutions or future work to address these limitations.

**Questions:**

Combine Figures 4, 5, and 6 into a single, more comprehensive figure
Add labels or annotations to highlight key features or differences between examples
Include a diverse set of objects to better showcase the method's capabilities

Comparision between the improvement in Part C and baseline method is missing.

---

> ### Author Response · Authors · 2024-11-25
> **Response to Reviewer z2rV**
>
> Thank you for your insightful and constructive comments! We have added additional experiments and modified our paper according to your comments.
>
> ### 1. The manuscript in Part C is too rough
> Thank you for pointing this out. We have provided a detailed explanation of the refinement step in General Response Section 4A, including mathematical equations and pseudocode for clarity. Additionally, we conducted a quantitative ablation study on the refinement step and included more visualizations comparing objects before and after refinement.
>
> In the quantitative ablation study, we test the following settings: 1. No refinement step applied. 2. Refinement step applied without random transformation (same as Richdreamer). 3. Refinement step applied with random transformation. The results are summarized in the table below.
>
>
> |            | No Refinement | Refinement w/o transformation | Refinement w/ transformation |
> | ---------- | ------------- | ----------------------------- | ---------------------------- |
> | CLIP Score | 0.7329        | 0.7928                        | **0.8205**                   |
> | VQA Score  | 0.6551        | 0.8164                        | **0.9376**                   |
>
> We also included additional qualitative results to illustrate the differences in geometry before and after refinement. These results can be viewed at the following link: https://drive.google.com/file/d/1Q7B2Z1WIocCE0saN2ggZvniGu3gDGO2q/view?usp=drive_link.
> For more details, please refer to General Response Section 3C and 3D.
>
> ### 2. Provide examples of cases where the pipeline fails or produces suboptimal results
> Visualizations of some failure cases can be found at the following link: https://drive.google.com/file/d/11l73OxPfDN9ZjT4GENErR2AeO1gHVFuR/view?usp=drive_link. In the first case, the refinement step mistakenly optimized a transparent glass door of a dishwasher and hallucinated dishes behind the door. This issue sometimes arises due to the randomness in the optimization process and the fact that the SDS loss for texture is computed in RGB space, which limits the albedo diffusion model’s understanding of explicit 3D structures. In the second case, inaccurate 3D segmentation of a plug resulted in artifacts. Since the current 3D segmentation step is not completely accurate, incorrect segmentation results can negatively affect subsequent steps. Therefore, Articulate Anything would benefit from a stronger open-vocabulary 3D segmentation model. Progress in segmentation models could significantly enhance the overall performance of Articulate Anything, addressing these limitations and reducing failure cases.
>
>
> ### Questions:
> > **Q1: Combine Figures 4, 5, and 6 into a single, more comprehensive figure Add labels or annotations to highlight key features or differences between examples Include a diverse set of objects to better showcase the method's capabilities.**
>
> Thank you for the advice. We have combined these figures and added captions to explain the results in the revised figure. Additionally, we have included more results from our pipeline in Appendix Sections B and E.
>
>
> > **Q2: Comparision between the improvement in Part C and baseline method is missing.**
>
> As our refinement step is developed over Richdreamer, we use it as the baseline for the refinement step. We report scores for the following settings: 1. No refinement step applied. 2. Refinement step applied without random transformation (same as Richdreamer). 3. Refinement step applied with random transformation. The results are summarized in the table below.
>
>
> |            | No Refinement | Refinement w/o transformation | Refinement w/ transformation |
> | ---------- | ------------- | ----------------------------- | ---------------------------- |
> | CLIP Score | 0.7329        | 0.7928                        | **0.8205**                   |
> | VQA Score  | 0.6551        | 0.8164                        | **0.9376**                   |
>
> For more details please refer to General Response Section 3C and 3D.
>
> *We wish that our response has addressed your concerns, and turns your assessment to the positive side. If you have any more questions, please feel free to let us know during the rebuttal window.*
>
> Best,
>
> Authors

---

### Official Review · Reviewer_uMgk · 2024-11-03

**Soundness:** 3
**Presentation:** 2
**Contribution:** 2
**Rating:** 5
**Confidence:** 3

**Summary:**

This paper addresses an interesting problem that aims to convert 3D meshes into articulated objects. This challenge is quite important and has the potential to greatly benefit the fields of 3D vision and robotics.

**Strengths:**

- The challenge discussed in this paper is important, and the proposed algorithm is reasonable.
- The 3D demos presented in this paper are interesting.

**Weaknesses:**

- Novelty: This paper introduces an interesting method called "Articulated Anything" to address the problem of articulated object generation. While the method is reasonable, it essentially relies on the power of various large models and diffusion models, which may limit the novelty of the proposed framework.

- Writing: Some parts of this paper are difficult to follow. For example, in Section 3.4, the process of refinement in the proposed architecture is hard to follow. When describing the method, it would be helpful to include some mathematical expressions or pseudocode to assist in explaining the approach.

- Experiments: In the ablation study, it is recommended to add more quantitative experiments to evaluate the performance of different components of the proposed framework. For instance, for the ablations of refinement and transformation presented in Figure 6, could the authors provide detailed quantitative comparisons for these experiments?

- Performance: The performance of the proposed method is not particularly impressive. It is difficult to observe a significant improvement compared to existing methods, such as CAGE.

**Questions:**

Please see the weaknesses part.

---

> ### Author Response · Authors · 2024-11-25
> **Response to Reviewer uMgk**
>
> Thank you for your insightful and constructive comments! We have added additional experiments and modified our paper according to your comments.
>
> ### 1. Novelty: While the method is reasonable, it essentially relies on the power of various large models and diffusion models, which may limit the novelty of the proposed framework.
>
> We use large models and diffusion models over existing pretrained models to leverage the vast knowledge embedded in these models for achieving open-vocabulary mesh-to-articulated-object conversion—a novel task that has not been addressed before.
>
> Effectively utilizing large models for complex tasks is non-trivial and requires innovation, as the process of extracting their knowledge is not straightforward. For example, you cannot simply input an image into GPT4o and expect per-pixel segmentation masks; instead, you need to pre-segment the image and label each part using techniques like SoM. Similarly, GPT4o cannot directly process a 3D mesh; instead, you must design visual prompts that accurately map back to the 3D domain without ambiguity. In conclusion, we believe that Articulate Anything demonstrates significant novelty in addressing these challenges.
>
> ### 2. Writing: Some parts of this paper are difficult to follow
> Thank you for pointing out the writing issues in our paper. We have provided a detailed explanation of the refinement step in General Response Section 4A, including mathematical equations and pseudocode to clarify the process. Additionally, the prompting details for the 3D segmentation and articulation estimation steps are included in Appendix Section F for further clarification.
>
> ### 3. Experiments: In the ablation study, it is recommended to add more quantitative experiments to evaluate the performance of different components of the proposed framework
> We compare three different settings to validate the effectiveness of refinement step: 1. No refinement step applied. 2. Refinement step applied without random transformation (same as Richdreamer). 3. Refinement step applied with random transformation. The results are presented in the table below. The highest scores are achieved when using refinement with random transformation, demonstrating the effectiveness of our refinement step.
>
> |            | No Refinement | Refinement w/o transformation | Refinement w/ transformation |
> | ---------- | ------------- | ----------------------------- | ---------------------------- |
> | CLIP Score | 0.7329        | 0.7928                        | **0.8205**                   |
> | VQA Score  | 0.6551        | 0.8164                        | **0.9376**                   |
>
>
> ### 4. Performance: The performance of the proposed method is not particularly impressive.
>
> It is worth noting that Articulate Anything generates articulated objects in an open-vocabulary manner, whereas NAP and CAGE are restricted to specific categories of articulated objects based on their training datasets. The experiments for articulation parameter estimation were conducted on object categories that NAP and CAGE were trained on. While the performance of Articulate Anything on these specific categories is comparable to CAGE, its range of operable categories is significantly broader. Unlike NAP and CAGE, which is not able to operate on unseen categories, Articulate Anything does not have such concern.
>
> For unconditional generation, CAGE represents parts using axis-aligned bounding boxes and subsequently retrieves parts from the PartNet-Mobility dataset. This approach imposes an additional limitation on CAGE, as the number of parts available in PartNet-Mobility is inherently restricted.
>
> The most significant advantage of Articulate Anything over existing methods is that it is an open-vocabulary approach that does not require any labeled articulated object data. It is capable of generalizing to a broad range of object categories, overcoming the limitations of category-specific models like NAP and CAGE.
>
> *We wish that our response has addressed your concerns, and turns your assessment to the positive side. If you have any more questions, please feel free to let us know during the rebuttal window.*
>
> Best,
>
> Authors

---

> > ### Comment · Reviewer_uMgk · 2024-11-26
> >
> > Thank you for your responses and for updating your paper with additional experiments. Overall, the paper has been improved during the rebuttal period and addresses some of my concerns. However, after reading your responses and the comments from other reviewers, I have decided to maintain my original rating. First, I agree with Reviewer rxUy regarding the need to provide more details about the merging mechanism and comparisons with the latest baselines. Secondly, the authors claim that the proposed method can generalize to unseen categories, whereas NAP and CAGE cannot. However, it would be more convincing to use experiments to support this claim. For instance, the authors could include additional experiments showing test results on new categories that did not appear in the training process of NAP and CAGE.

---

> > > ### Author Response · Authors · 2024-11-29
> > >
> > > Thank you for your constructive feedback. In response to your concerns, we have conducted additional experiments and provided further details.
> > >
> > > > **First, I agree with Reviewer rxUy regarding the need to provide more details about the merging mechanism**
> > >
> > > Our pipeline incorporates two merging steps to process 2D masks generated by SAM into 3D masks at the granularity of actual parts.
> > >
> > > The first merging step merges 2D masks produced by SAM into 3D masks according to overlap ratio. Given two 2D masks, $M_A, M_B$ from different views, we first project $M_A$ onto the view of $M_B$ and compute the overlap ratio as $\frac{NUMBER\ \ OF\ \  OVERLAP\ \  PIXELS}{NUMBER\ \ OF\ \ PIXELS\ \  IN\ \  M_A}$. The process is then repeated by projecting $M_B$ onto the view of $M_A$. If both overlap ratios exceed the predefined threshold, the two masks are merged. For all our experiments, we set this hyperparameter to 0.4.
> > >
> > > The second step merges adjacent 3D masks with the same semantic label. Specifically, two adjacent 3D masks $M_1, M_2$ with the same semantics label are merged unless there exists at least one pair of 2D masks $m_1, m_2$, where $m_1$ is a 2D mask component of $M_1$, $m_2$ is a 2D mask component of $M_2$, such that $m_1$ and $m_2$ are two different instances of the same semantic part label, and they co-occur in at least one image.
> > >
> > > These two merging steps together help to refine the over-segmented masks produced by SAM into meaningful parts. **For more details, please refer to Appendix Section F Prompting Details, paragraph 3D Segmentation.**
> > >
> > > > **Comparisons with the latest baselines.**
> > >
> > >
> > > We have additionally compared the 3D segmentation performance of Articulate Anything with PartSlip++. The results, shown in the table below, indicate that Articulate Anything achieves the best overall performance in 3D segmentation.
> > >
> > > |   Method   | Overall (mIOU) | Bottle |   Chair   |  Display  |   Door    | Knife | Lamp      | StorageFurniture |   Table   | Camera |   Cart    | Dispenser |  Kettle   | KitchenPot | Oven      | Suitcase |  Toaster  |
> > > |:----------:|:--------------:|:------:|:---------:|:---------:|:---------:|:-----:|:--------- |:----------------:|:---------:|:------:|:---------:|:---------:|:---------:|:----------:|:--------- |:--------:|:---------:|
> > > | PartSLIP++ |     44.21      | 65.42  | **77.67** | **76.36** |   42.58   | 5.76  | 38.36     |      35.32       |   29.21   | **48.67**  |   81.16   |   6.54    |   32.06   | **80.95**  | 26.30     |  **42.75**  |   18.24   |
> > > |    Ours    |   **51.67**    | **73.44**  |   71.22   |   72.77   | **56.03** | **36.82** | **72.29** |    **46.10**     | **46.69** | 31.13  | **84.34** | **17.26** | **71.85** |   58.41    | **32.05** |  33.13   | **23.13** |
> > >
> > > We also compared Articulate Anything with OPDFormer for articulation parameter estimation. Using the official OPDMulti GitHub repository, we retrained OPDFormer on the "Onedoor" dataset, which was also used to train ANCSH and OPD. The results are presented below, showing that Articulate Anything outperformed OPDFormer.
> > >
> > > |                          | ANCSH |  OPD  | OPDFormer |   Ours    |
> > > | ------------------------ |:-----:|:-----:|:---------:|:---------:|
> > > | error in Joint direction | 6.74  | 10.73 |   9.66    | **5.37**  |
> > > | error in Joint position  | 0.065 | 0.117 |   0.108   | **0.049** |
> > >
> > >
> > > > **the authors could include additional experiments showing test results on new categories that did not appear in the training process of NAP and CAGE.**
> > >
> > > We conducted experiments to test the generalizability of NAP, CAGE, and Articulate Anything on object categories in PartNet-Mobility that are not part of CAGE's training set (e.g., laptop, cart, door, etc.). The input to each method remains consistent with the previous articulation parameter estimation experiment:
> > >
> > > * **NAP**: Ground truth vertices (e.g., part bounding boxes, spatial locations, shape latents) are provided, while edges (joint parameters) are estimated.
> > > * **CAGE**: Attributes such as bounding boxes, joint types, and semantic labels for each node are provided, while joint axes and ranges are estimated.
> > > * **Articulate Anything**: Shapes, part semantics, and joint types are provided, and the joint axes and limits are estimated.
> > >
> > > Errors are measured as follows:
> > >
> > > * **Joint direction error**: The angle between the ground truth and predicted axis.
> > > * **Joint position error**: The distance between the ground truth axis and the predicted axis.
> > >
> > > The results are shown in the table below:
> > >
> > > |                          |  NAP  | CAGE  |   Ours    |
> > > | ------------------------ |:-----:|:-----:|:---------:|
> > > | error in Joint direction | 42.23  | 58.64 | **4.81**  |
> > > | error in Joint position  | 0.225 | 0.192 | **0.075** |
> > >
> > > The results indicate that the performance of NAP and CAGE drastically degrades on unseen object categories, whereas Articulate Anything maintains performance comparable to the categories in CAGE's training set.

---

### Official Review · Reviewer_rxUy · 2024-11-04

**Soundness:** 2
**Presentation:** 1
**Contribution:** 2
**Rating:** 5
**Confidence:** 5

**Summary:**

This paper presents a pipeline for part segmentation, motion prediction, part completion with re-texturing. The authors leverage vision foundation models and LLM to lift the 2D part segmentation to 3D. The authors further design the heuristic-based motion prediction and leverage LLM to pick the keypoint for the predefined joint categories. For the incomplete geometry, they use diffusion priors to optimize the geometry and texture with text prompts. They compare their results with some articulated model generative work.

**Strengths:**

1. The task to convert static 3D mesh into interactable articulated objects is valuable and interesting.
2. The final demos that show that it can guide the real-to-sim-to-real demonstrate the usefulness of the pipeline.

**Weaknesses:**

1. This paper is not an articulated object generation work. But focus on the part segmentation, motion prediction and part completion. It’s more on the analyzing on the shape side instead of generating the shape. Therefore, the whole comparison is a bit weird. There are a number of works focusing on part segmentation, and motion prediction. There needs to be comparisons with them on the segmentation and motion prediction performance (e.g. Category-Level Articulated Object Pose Estimation, PartSLIP). The authors can consider comparing with some work listed in the survey (Survey on Modeling of Human-made Articulated Objects).
2. A known issue of lifting 2D SAM masks into 3D is the consistency of different viewpoints, the granularity of the segmentation, and how to predefine the part labels. There lacks such discussion in the paper. Please provide more discussions on such details and show more quantative results on this part.
3. The writing and experiments are confusing to mention generated objects from the pipeline. When compared with other generative work, it’s unclear if you start from an existing 3D mesh. If yes, then why compare the raw mesh with them, even if there can be some geometry change in the part completion step. Please provide a more detailed explanation on the input of different methods of the comparison and explain why the comparison is fair if the proposed method uses an existing 3D mesh. Please also discuss how to fetch the 3D mesh to compare with other methods.

**Questions:**

For the shape in the teaser image and in the website, do they go through the refinement step? Seems that their geometry is much better than the results after the refinement step in the paper. Please give more clarification on the process to generate the demos of the objects shown in the teaser image on how do they keep the original texture, if they go through the whole pipeline of the proposed method?

---

> ### Author Response · Authors · 2024-11-25
> **Response to Reviewer rxUy [1/2]**
>
> Thank you for your insightful and constructive comments! We have added additional experiments and modified our paper according to your comments.
>
> ### 1. This paper is not an articulated object generation work.... There needs to be comparisons with them on the segmentation and motion prediction performance (e.g. Category-Level Articulated Object Pose Estimation, PartSLIP).
>
> As stated in General Response Section 2, we sincerely appreciate your feedback in helping us clarify the primary task of our pipeline, and we have revised our paper accordingly. Previously, we regarded our pipeline as generative because, by integrating an exisitng 3D generation model to generate surface meshes as the input to Articulate Anything, our pipeline takes advanatge of the open-vocabulary 3D generation capabilities of the 3D generation model and inherits its generative paradigm. Additionally, we aimed to evaluate the performance of Articulate Anything in an end-to-end manner and compare it with other state-of-the-art methods. After the reform, we no longer consider Articulate Anything as a generative framework. Instead, we view open-vocabulary articulated object generation as a novel downstream task enabled by our pipeline. Under this perspective, we included quantitative experiments to evaluate the performance of 3D segmentation and articulation parameter estimation, comparing against other open-vocabulary 3D segmentation methods and motion prediction works.
>
> For 3D segmentation, we use PartSlip and PartDistill as baselines and conduct evaluation on the PartNetE dataset. The results are presented in the table below. For details regarding the experimental setup, please refer to General Response Section 3A.
>
> |   Method    |  Overall (mIOU)  |  Bottle   |   Chair   |  Display  |   Door    |   Knife   | Lamp      | StorageFurniture |   Table   |  Camera   |   Cart    | Dispenser |  Kettle   | KitchenPot | Oven      | Suitcase  |  Toaster  |
> |:-----------:|:---------:|:---------:|:---------:|:---------:|:---------:|:---------:|:--------- |:----------------:|:---------:|:---------:|:---------:|:---------:|:---------:|:----------:|:--------- |:---------:|:---------:|
> |  PartSLIP   |   38.83   |   76.12   | **73.85** |   59.10   |   14.57   |   10.04   | 43.20     |      27.30       |   33.24   | **51.40** |   79.54   |   10.22   |   22.57   |   31.67    | 31.08     |   42.58   |   14.76   |
> | PartDistill |   42.98   | **77.98** |   70.23   |   60.79   |   43.86   | **39.41** | 67.56     |      21.60       |   42.04   |   32.75   |   76.39   |   11.45   |   36.92   |   21.44    | 27.89     | **45.36** |   10.52   |
> |    Ours     | **51.67** |   73.44   |   71.22   | **72.77** | **56.03** |   36.82   | **72.29** |    **46.10**     | **46.69** |   31.13   | **84.34** | **17.26** | **71.85** | **58.41**  | **32.05** |   33.13   | **23.13** |
>
> For articulation parameter estimation, we initially compared against NAP and CAGE, as these two generative methods can produce joint configurations conditioned on complete part geometries, which closely aligns with the input to our articulation parameter estimation step. We now additionally include ANCSH and OPD as baselines, using single-observation RGB-D input. The results are reported in the table below.
> |                          | ANCSH | OPD   | Ours  |
> | ------------------------ | ----- |:----- |:----- |
> | error in Joint direction | 6.74  | 10.73 | **5.37** |
> | error in Joint position  | 0.065 | 0.117 | **0.049** |
>
> Experimental results show that Artiulate Anything performs better than NAP and CAGE. For details of the experimental setup, please refer to General Response section 3B.

---

> ### Author Response · Authors · 2024-11-25
> **Response to Reviewer rxUy [2/2]**
>
> ### 2. Please provide more discussions on such details and show more quantative results on this part.
>
> We observed that SAM provides inconsistent segmentation across different views, along with varying granularity. Additionally, SAM tends to over-segment, with segmentation granularity that is typically finer than that of a semantic part. Leveraging this observation, we designed two merging steps to transform the 2D segmentation masks from SAM into 3D segmentation masks of semantic parts.
>
> The first merging step merges 2D segmentation masks across different views into 3D segmentation masks based on overlap ratios. The semantics of each 3D mask are determined by the most frequent semantic label among its 2D mask components.
>
> The second merging step further combines adjacent 3D segmentation masks with the same semantics. To prevent merging different instances of the same semantic part (e.g., two adjacent drawers), we prompt GPT4o to differentiate instances within the same semantic category.
>
> Part labels are generated using GPT4o. We simply provide rendered RGB images as input to GPT4o and ask it to identify semantic parts.
>
> For quantitative evaluation, we conducted experiments to compare the segmentation step of Articulate Anything with PartSlip and PartDistill. Results in the previous table clearly demonstrate our advantage over both baselines. Details are provided in General Response Section 3A. These results further validate that by properly utilizing large foundation models like SAM and GPT4o, we achieve superior segmentation performance compared to methods relying on less capable pretrained models.
>
> ### 3. Please provide a more detailed explanation on the input of different methods of the comparison and explain why the comparison is fair if the proposed method uses an existing 3D mesh. Please also discuss how to fetch the 3D mesh to compare with other methods.
> In the unconditional generation experiment, all input meshes for Articulate Anything are generated using InstantMesh, an image-to-3D generative model. The input images for InstantMesh are created using Stable Diffusion, conditioned on text prompts corresponding to object category names. We do not use existing meshes from datasets or websites for the unconditional generation experiments.
> In contrast, NAP and CAGE source their parts from the PartNet-Mobility dataset, which consists of pre-existing meshes. Therefore, we believe this comparison is fair, if not slightly biased in favor of NAP and CAGE.
>
> ### Questions
> > **Q1: For the shape in the teaser image and in the website, do they go through the refinement step? Seems that their geometry is much better than the results after the refinement step in the paper. Please give more clarification on the process to generate the demos of the objects shown in the teaser image on how do they keep the original texture, if they go through the whole pipeline of the proposed method?**
>
> The objects featured in our teaser are sourced from Objaverse (as metioned in the caption of figure 1) and do not undergo the refinement step. As described in the "Annotate 3D Object Datasets" section of the Applications, we only apply the 3D segmentation and articulation estimation steps of our pipeline to annotate objects retrieved from Objaverse. TThe refinement step is specifically designed for surface meshes with incomplete part geometry. Given the current limitations of recent 3D generation methods (e.g., SDS optimization and others), their output still falls significantly short of artist-crafted meshes. Consequently, it is not optimal to apply the refinement step to artist-crafted meshes, particularly those with complete part geometry and inner structure.
>
> *We wish that our response has addressed your concerns, and turns your assessment to the positive side. If you have any more questions, please feel free to let us know during the rebuttal window.*
>
> Best,
>
> Authors

---

> > ### Comment · Reviewer_rxUy · 2024-11-25
> >
> > Thanks for the detailed responses and additional experiments from the authors. The paper improves a lot with a more clear task setting and comparisons with work in each module. However, there can still be some more improvements to make the work better. For the 2D lifting part, the merging strategy is always very sensitive to the hyperparameter choices (e.g., iou threhsold), especially in the part segmentation setting where all parts are very close. And the authors metion that the GPT-4o is triggered to further help determine if two parts should be merged. There can be more details about how to choose the proper image with the proper viewpoint to trigger GPT-4o, and is there some triggering requirement to trigger it. It's great to see the additional essential experiments, but the work can be more solid if comparing to more close-date method, like PartSlip++ and OPDFormer, which are the more recent version of PartSlip and OPD). I will raise my score to marginally below, and I feel that the paper can be much more solid with more improvements.

---

> > > ### Author Response · Authors · 2024-11-29
> > >
> > > Thank you for your constructive feedback. In response to your concerns, we have conducted additional experiments and provided further details.
> > >
> > > > **For the 2D lifting part, the merging strategy is always very sensitive to the hyperparameter choices (e.g., iou threhsold), especially in the part segmentation setting where all parts are very close.**
> > >
> > > A predefined hyperparameter is required for the first merging step (merging 2D masks produced by SAM into 3D masks). Given two 2D masks, $M_A, M_B$ from different views, we first project $M_A$ onto the view of $M_B$ and compute the overlap ratio as $\frac{NUMBER\ \ OF\ \  OVERLAP\ \  PIXELS}{NUMBER\ \ OF\ \ PIXELS\ \  IN\ \  M_A}$. The process is then repeated by projecting $M_B$ onto the view of $M_A$. If both overlap ratios exceed the predefined threshold, the two masks are merged. For all our experiments, we set this hyperparameter to 0.4, which worked fine. We also find that slightly adjusting this hyperparameter (from 0.3 to 0.5 for example) doesn't influence much on the performance.
> > >
> > > > **And the authors metion that the GPT-4o is triggered to further help determine if two parts should be merged. There can be more details about how to choose the proper image with the proper viewpoint to trigger GPT-4o, and is there some triggering requirement to trigger it.**
> > >
> > > The answer is that we do not select a specific viewpoint because every rendered image (16 in total for a single object) undergoes this process. There is no triggering requirement; instead, it is up to GPT4o to determine whether there are multiple instances of the same semantic part label.
> > >
> > > The detailed description is provided below.
> > >
> > > In the first merging step, GPT does not influence the merging process.
> > >
> > > In the second merging step, two adjacent 3D masks $M_1, M_2$ with the same semantics label are merged unless there exists at least one pair of 2D masks $m_1, m_2$, where $m_1$ is a 2D mask component of $M_1$, $m_2$ is a 2D mask component of $M_2$, such that $m_1$ and $m_2$ are two different instances of the same semantic part label, and they co-occur in at least one image. GPT4o aids this process by determining which 2D masks generated by SAM in the same 2D image correspond to different instances of the same semantic part label.
> > >
> > > After applying SAM to all rendered 2D images and labeling them with Set-of-Mark techniques (the 2D masks are annotated on the image, with a numeric label placed at the center of each mask), the labeled images are fed into GPT4o, which is prompted to assign the masks to semantic parts. (at this stage, we already know the movable parts of interest.) The prompt instructs GPT4o to distinguish between different instances of the same semantic part label while assigning 2D masks to semantic parts. **For more details, refer to Appendix Section F: Prompting Details.**
> > >
> > >
> > > > **It's great to see the additional essential experiments, but the work can be more solid if comparing to more close-date method, like PartSlip++ and OPDFormer, which are the more recent version of PartSlip and OPD.**
> > >
> > > We have additionally compared the 3D segmentation performance of Articulate Anything with PartSlip++, using the official implementation from the PartSlip2 GitHub repository. The input for PartSlip++ is the same as PartSlip. The results, shown in the table below, indicate that Articulate Anything achieves the best overall performance in 3D segmentation. (The overall mIoU for PartSLIP and PartDistill are 38.83 and 42.98, respectively.)
> > >
> > > |   Method   | Overall (mIOU) | Bottle |   Chair   |  Display  |   Door    | Knife | Lamp      | StorageFurniture |   Table   | Camera |   Cart    | Dispenser |  Kettle   | KitchenPot | Oven      | Suitcase |  Toaster  |
> > > |:----------:|:--------------:|:------:|:---------:|:---------:|:---------:|:-----:|:--------- |:----------------:|:---------:|:------:|:---------:|:---------:|:---------:|:----------:|:--------- |:--------:|:---------:|
> > > | PartSLIP++ |     44.21      | 65.42  | **77.67** | **76.36** |   42.58   | 5.76  | 38.36     |      35.32       |   29.21   | **48.67**  |   81.16   |   6.54    |   32.06   | **80.95**  | 26.30     |  **42.75**  |   18.24   |
> > > |    Ours    |   **51.67**    | **73.44**  |   71.22   |   72.77   | **56.03** | **36.82** | **72.29** |    **46.10**     | **46.69** | 31.13  | **84.34** | **17.26** | **71.85** |   58.41    | **32.05** |  33.13   | **23.13** |
> > >
> > > We also compared Articulate Anything with OPDFormer for articulation parameter estimation. Using the official OPDMulti GitHub repository, we retrained OPDFormer on the "Onedoor" dataset, which was also used to train ANCSH and OPD. The results are presented below, showing that Articulate Anything outperformed OPDFormer.
> > >
> > > |                          | ANCSH |  OPD  | OPDFormer |   Ours    |
> > > | ------------------------ |:-----:|:-----:|:---------:|:---------:|
> > > | error in Joint direction | 6.74  | 10.73 |   9.66    | **5.37**  |
> > > | error in Joint position  | 0.065 | 0.117 |   0.108   | **0.049** |

---

### Author Response · Authors · 2024-11-25
**General Response [1/5]**

Thank you for your insightful and constructive comments! We have added additional experiments and modified our paper according to your comments.

**1. Our Contributions** We are pleased that the reviewers have generally recognized our contributions:

* We proposed an important challenge of converting rigid meshes into their articulated counterparts.
* We introduced a novel pipeline to tackle this challenge and demonstrated promising results.
* We developed an articulation parameter estimation method based on heuristic rules and visual prompting.

**2. Paper Reorganization** We sincerely thank all reviewers for pointing out areas in our paper that may cause misunderstandings and highlighting inconsistencies between the task our pipeline addresses and the experimental setup. We acknowledge that the primary goal of Articulate Anything is to convert a rigid mesh into its articulated counterpart, and it should not be regarded as a generative pipeline. Instead, articulated object generation represents an important and practical downstream application, achieved by utilizing an existing 3D generation model to produce surface meshes as input for Articulate Anything.
To address these concerns, we have updated the [pipeline figure](https://drive.google.com/file/d/1BKCpzM61AfEec79JQH-sgS5zM430DkAl/view?usp=drive_link) and revised the experimental section to ensure consistency throughout the paper. The parts we have modified in our paper are highlighted. Additionally, we included new quantitative and qualitative experiments focusing on 3D segmentation and articulation parameter estimation, making our paper self-contained. Details of these updates are provided in the following sections.

We have changed the title to ARTICULATE ANYTHING: OPEN-VOCABULARY 3D ARTICULATED OBJECTS MODELING to avoid potential confusion. Additionally, we updated the appendix to include implementation details, clarifications on experimental settings, and results from additional experiments.

---

> ### Author Response · Authors · 2024-11-25
> **General Response [2/5]**
>
> **3. New Experiments**
>
> * **[A] Quantitative comparison for 3D Segmentation.** We compare the semantic segmentation performance of the segmentation component in Articulate Anything to the zero-shot version of PartSlip[1] and PartDistill[5]. The comparison focuses on object categories from the PartnetE dataset proposed by PartSlip. Each object category in PartNetE has predefined part labels, and we adhere to this setup, evaluating segmentation performance exclusively on these predefined parts. For PartSlip and PartDistill, the input is a multi-view fusion point cloud generated by projecting RGB-D pixels from rendered images into 3D space, and the output is a semantic label for each point in the input point cloud. For Articulate Anything, the input is a 3D surface mesh. It then undergoes the 3D segmentation step, producing a labeled point cloud as output. The predicted point cloud segmentation is then compared with the ground-truth segmentation, and mIOU is measured. The results, shown in the table below, demonstrate that the segmentation component of Articulate Anything outperforms PartSlip and PartDistill overall.
>
> |   Method    |  Overall (mIOU)  |  Bottle   |   Chair   |  Display  |   Door    |   Knife   | Lamp      | StorageFurniture |   Table   |  Camera   |   Cart    | Dispenser |  Kettle   | KitchenPot | Oven      | Suitcase  |  Toaster  |
> |:-----------:|:---------:|:---------:|:---------:|:---------:|:---------:|:---------:|:--------- |:----------------:|:---------:|:---------:|:---------:|:---------:|:---------:|:----------:|:--------- |:---------:|:---------:|
> |  PartSLIP   |   38.83   |   76.12   | **73.85** |   59.10   |   14.57   |   10.04   | 43.20     |      27.30       |   33.24   | **51.40** |   79.54   |   10.22   |   22.57   |   31.67    | 31.08     |   42.58   |   14.76   |
> | PartDistill |   42.98   | **77.98** |   70.23   |   60.79   |   43.86   | **39.41** | 67.56     |      21.60       |   42.04   |   32.75   |   76.39   |   11.45   |   36.92   |   21.44    | 27.89     | **45.36** |   10.52   |
> |    Ours     | **51.67** |   73.44   |   71.22   | **72.77** | **56.03** |   36.82   | **72.29** |    **46.10**     | **46.69** |   31.13   | **84.34** | **17.26** | **71.85** | **58.41**  | **32.05** |   33.13   | **23.13** |
>
> * **[B] Articulation Parameters Estimation.** We compare Articulate Anything with other methods focused on articulation estimation. For OPD[2], we use the official GitHub repository and the RGB-D input checkpoint. For ANCSH[3], we use the PyTorch implementation provided by OPD's authors. The evaluation is conducted on the "Onedoor" dataset, a subset of OPDsynth, which includes objects from various categories in PartNet-Mobility that feature doors. This dataset is used to train ANCSH in its pytorch implementation. ANCSH uses a single-view point cloud as input, while OPD uses an RGB image (with optional depth information). Both methods output the segmentation of relevant parts and their joint parameters. To ensure a fair comparison, our pipeline is adapted to the single-observation RGB-D setting: only one image (the input observation) undergoes segmentation, and the results are directly projected as a point cloud with segmentation labels. This single-view point cloud is then processed in the second step for articulation parameter estimation. The results, shown in the table below, demonstrate that Articulate Anything outperforms both ANCSH and OPD, even when evaluated in-domain.
>
> |                          | ANCSH | OPD   | Ours  |
> | ------------------------ | ----- |:----- |:----- |
> | error in Joint direction | 6.74  | 10.73 | **5.37** |
> | error in Joint position  | 0.065 | 0.117 | **0.049** |
>
>
> * **[C] Quantitative Ablation Study for Refinement Step.** In the ablation study, we evaluate the same objects used in the quantitative evaluation of unconditional generation. We apply three different settings to the intermediate output of the articulation estimation step: 1. No refinement step applied. 2. Refinement step applied without random transformation (same as Richdreamer). 3. Refinement step applied with random transformation. The results are presented in the table below. The highest scores are achieved when using refinement with random transformation, demonstrating the effectiveness of our refinement step.
>
> |            | No Refinement | Refinement w/o transformation | Refinement w/ transformation |
> | ---------- | ------------- | ----------------------------- | ---------------------------- |
> | CLIP Score | 0.7329        | 0.7928                        | **0.8205**                   |
> | VQA Score  | 0.6551        | 0.8164                        | **0.9376**                   |

---

> ### Author Response · Authors · 2024-11-25
> **General Response [3/5]**
>
> * **[D] Qualitative Ablation Study for Refinement Step.** We also provide visualizations of the input and output of the refinement step, as shown in the link: https://drive.google.com/file/d/1Q7B2Z1WIocCE0saN2ggZvniGu3gDGO2q/view?usp=drive_link. The pencil case and lighter in the second and third rows feature manually set joint parameters and shape primitives as link geometries. The refinement step successfully optimizes storage space for the drawer and pencil case and generates a nozzle structure for the lighter, and produces plausible textures.

---

> ### Author Response · Authors · 2024-11-25
> **General Response [4/5]**
>
> **4. Implementation Details**
> * **[A] The Refinement Step.** In Richdreamer[4], the Score Distillation Sampling process proposed by DreamFusion is formulated as follows: Given a 3D representation $\phi$, and a differentiable renderer $g$, the rendered image is $x = g(\phi)$. The SDS loss is then used to optimize the 3D representation $\phi$: $\nabla_{\phi} \mathcal{L}\_{\text{SDS}}(\phi, x=g(\phi)) = \mathbb{E}\_{t, \epsilon}\left[w(t)(\epsilon\_{\theta}(z\_t ; y, t) - \epsilon) \frac{\partial x}{\partial \phi}\right]$, where $z_t$ is the noisy latent code, $\epsilon$ is the injected noise and $\epsilon_\theta$ is the noise predicted by a denoising model $\theta$, conditioned on timestep $t$ and text embedding $y$. The term $w(t)$ is a timestep dependent weighting factor. In Richdreamer and other previous works the 3D representation $\phi$ is static, whereas in our case, it is articulated. Omitting other attributes, we denote the 3D representation of articulated objects as $\phi_q$, where $q$ is a vector representing joint positions. During optimization, the base of the articulated object remains fixed. Since non-fixed joints of interest (revolute, prismatic, and continuous) all have one degree of freedom, each element in $q$ corresponds to the position of a non-fixed joint in $\phi_q$. The articulated object in its rest configuration is denoted as $\phi_{q_0}$. A transformation function $T$ maps $\phi_{q_0}$ to $\phi_q = T(\phi_{q_0}, q)$ given the desired joint positions $q$. The optimization process for Richdreamer and other previous SDS optimization methods can be briefly summarized by the following pseudo-code:
> ```text
> FOR i IN iterations:
>     render an image x (x = g(phi))
>     sample timestep t
>     compute SDS loss
>     update optimizer
> ```
>
> In our approach, we extend this process by sampling joint positions and transforming object parts accordingly:
> ```text
> FOR i IN iterations:
>     sample joint position q
>     transform parts according to q (phi_q = T(phi_q0, q))
>     render an image x (x = g(phi_q))
>     sample timestep t
>     compute SDS loss
>     update optimizer
> ```
> * **[B] Experiment Setup.**
> **Articulation parameter estimation:** We use objects from CAGE's test split. In our setup, the shapes of the testing objects are known, and articulation parameters are predicted. For NAP, ground truth vertices (e.g., part bounding boxes, spatial locations, shape latents) are provided, while edges (joint parameters) are estimated.  Since NAP uniformly represents all joints using Plücker coordinates, we evaluate only the translational component for prismatic joints and the rotational component for revolute joints. For CAGE, some attributes of each node, such as bounding boxes, joint types, and semantic labels, are provided, while others, including joint axes and ranges, are estimated. In Articulate Anything, the shapes, semantics of each part, and joint types are given, and the joint axes and limits are estimated.
> **Unconditional Generation:** We generate articulated objects using minimal input. For NAP, no conditions are provided; its diffusion model generates objects unconditionally. After generation, the initial shape is decoded from the generated shape latent and replaced by the nearest matching part mesh. For CAGE, a random articulated object from PartNet-Mobility, in CAGE's test split, is retrieved. Its connectivity graph and object category label are used as input to CAGE's diffusion model. After generating bounding boxes, part semantics, and joint parameters, part meshes are retrieved using CAGE's retrieval method. For URDFormer, an object category is randomly sampled, and an image is generated using Stable Diffusion 3 (prompted to produce a front-view image of an object in the sampled category with a white background). URDFormer then takes the generated image as input and outputs a URDF. In Articulate Anything, an image is generated similarly, followed by mesh generation using InstantMesh. The generated mesh is processed through the full pipeline of Articulate Anything to produce an articulated object. For PartNet-Mobility, objects in the relevant categories are randomly retrieved. For each generated object, we render eight views surrounding it and compute the CLIPScore and VQAScore. The input text for these scores is "a OBJECT\_CATEGORY".
>
> * **[C] Prompting Details.** Please refer to appendix section G prompting details.

---

> ### Author Response · Authors · 2024-11-25
> **General Response [5/5]**
>
> [1] Liu, Minghua, et al. "Partslip: Low-shot part segmentation for 3d point clouds via pretrained image-language models." Proceedings of the IEEE/CVF conference on computer vision and pattern recognition. 2023.
>
> [2] Jiang, Hanxiao, et al. "OPD: Single-view 3D openable part detection." European Conference on Computer Vision. Cham: Springer Nature Switzerland, 2022.
>
> [3] Li, Xiaolong, et al. "Category-level articulated object pose estimation." Proceedings of the IEEE/CVF conference on computer vision and pattern recognition. 2020.\
>
> [4] Jiang, Hanxiao, et al. "OPD: Single-view 3D openable part detection." European Conference on Computer Vision. Cham: Springer Nature Switzerland, 2022.
>
> [5] Umam, Ardian, et al. "PartDistill: 3D Shape Part Segmentation by Vision-Language Model Distillation." Proceedings of the IEEE/CVF Conference on Computer Vision and Pattern Recognition. 2024.

---

### Meta-Review · Area_Chair_y6gR · 2024-12-23

**Metareview:**

**Summary**

The paper proposes a pipeline to take a 3D mesh, and produce an articulated version of the mesh, through three stages: 1) movable part segmentation, 2) articulation estimation, and 3) refinement.
Much of the pipeline relies on combinining recent advances with prompting GPT4o: part segmentation (use Part123 with SAM + set-of-marks prompting of GPT4o for labels), articulation prediction (identifying connected areas and prompting GPT4o), use geometry generation with SDS loss for refinement.  Experiments compare the performance of the different stages on PartNet-Mobility, and qualitative examples are shown for objects from Objaverse.

**Strengths**

1. Automatic creation of articulated objects is a important problem [uMgk,E2pY]
2. Proposed pipeline is reasonable [uMgk,z2rV,E2pY]

**Weaknesses**

1. Framing of the work [rxUy,E2pY]
   - Whether the work presented is a generative model or a model that analyzes an input mesh to create an articulated mesh
   - What precisely is the input to the proposed pipeline (input mesh as depicted in Figure 2, or single view image / nothing as demonstrated in experiments)
   - How much of the detailed geometry of the input mesh is actually preserved.
2. Concerns about experimental setup and validity [rxUy,uMgk,E2pY]
   - Lack of comparison with prior work and limited ablations
   - Whether comparing articulation parameter estimation for against generative models such as NAP and CAGE is appropriate. It seems more appropriate to compare against methods that are aims to predict articulation parameter given an input (vs generating different distributions of articulation parameters)
   - Whether the evaluation of generated visual quality using CLIP-score and VQA-score is meaningful for unconditional generation.
3. Lack of details and clarity [rxUy,uMgk,E2pY,z2rV]

**Recommendation**

Reviewers were slightly negative on the work, mostly due to issues with problem framing, lack of clarity, and important comparisons in the initial version.  While the manuscript has been updated, the reviewers still had their doubts about the validity of the experiments.

The AC finds the problem of creating 3D articulated assets (either from existing mesh or single view image or unconditioned) to be an important problem.  However, the AC shares the concerns of the reviewers whether appropriate evaluation and comparisons were performed to understand the performance and limitations of the proposed approach.  Due to the concerns expressed by reviewers, the AC finds the work not ready for publication at ICLR 2025.

**Additional Comments On Reviewer Discussion:**

The paper initially received divergent scores: 3 [rxUy],5 [E2pY],5 [uMgk], 8 [z2rV].  Reviewers had concerns about the framing of the proposed approach (e.g. was the work generating new 3D objects using a generative model, or performing analysis on an existing mesh or doing reconstruction from a single view image), poor experimental setup and lack of comparisons, as well as lack of details in the submission.

The positive reviewer [z2rV] found the proposed pipeline to be effective and liked the goal of open-vocabulary generation.  Like the other reviewers, z2rV also found parts of the paper rough and unclear with lack of comparison and analysis.  During the author response period, the authors made considerable revisions to the paper to try to address the reviewer concerns (e.g. adding experiments and providing more details in the appendix).  The AC note that as not all revisions to the manuscript was highlighted, it was also difficult for reviewers to identify what was updated.

However, reviewers remained mostly unconvinced.  The most negative reviewer [rxUy] increase their rating to 5, but the most positive one [z2rV] decreased their rating to 6.  Reviewer E2pY indicated that the paper as it currently stands still does not make clear the assumptions, making it difficult to judge the validity of the experiments.  Despite the updates, the AC agrees it was difficult to judge the validity of the experiments.

---

### Decision · Program_Chairs · 2025-01-22

Reject